




**Carbon dioxide emissions from the flat bottom and shallow**
**Nam Theun 2 Reservoir: drawdown area as a neglected**
**pathway to the atmosphere**
**Chandrashekhar Deshmukh[1,2,3,a], Frédéric Guérin[1,4,5], Axay Vongkhamsao[6],**
**Sylvie Pighini[6,b], Phetdala Oudone[6,c], Saysoulinthone Sopraseuth[6], Aranud**
**Godon[6,d], Wanidoporn Rode[6], Pierre Guédant[6], Priscia Oliva[1], Stéphane Audry[1],**
**Cyril Zouiten[1], Corinne Galy-Lacaux[2], Henri Robain[7], Olivier Ribolzi[1], Arun**
**Kansal[3], Vincent Chanudet[8], Stéphane Descloux[8], Dominique Serça[2]**
[1]{Géosciences Environnement Toulouse (GET), Université de Toulouse (UPS), 14 Avenue
E. Belin, F-31400 Toulouse, France}
[2]{Laboratoire d'Aérologie - Université de Toulouse - CNRS UMR 5560; 14 Av. Edouard
Belin, F-31400, Toulouse, France}
[3]{Centre for Regulatory and Policy Research, TERI University, New Delhi, India}
[4]{IRD ; UR 234, GET ; 14 Avenue E. Belin, F-31400, Toulouse, France}
[5]{Departamento de Geoquimica, Universidade Federal Fluminense, Niteroi-RJ, Brasil}
[6]{Nam Theun 2 Power Company Limited (NTPC), Environment & Social Division – Water
Quality and Biodiversity Dept.– Gnommalath Office, PO Box 5862, Vientiane, Lao PDR}
[7]{IRD, iEES-Paris, Centre IRD France-Nord, 32 avenue Henri Varagnat, 93143 Bondy
Cedex, France}
[8]{Electricité de France, Hydro Engineering Centre, Sustainable Development Dpt, Savoie
Technolac, F-73373 Le Bourget du Lac, France}
[a]{now at: Asia Pacific Resources International Limited (APRIL), Indonesia}
[b]{now at: Innsbruck University, Institute of Ecology, 15 Sternwartestrasse, A-6020
Innsbruck, Austria and Foundation Edmund Mach, FOXLAB-FEM, Via E. Mach 1, IT-38010
San Michele all'Adige, Italy}
[c]{now at: Department of Environmental Science, Faculty of Environmental Sciences,
National University of Laos, Vientiane, Lao PDR}





[d]{now at: Arnaud Godon Company, 44 Route de Genas, Nomade Lyon, 69003 Lyon, France}
Correspondence to: F. Guérin (Frederic.guerin@ird.fr)
**Abstract**
Freshwater reservoirs are a significant source of $CO_2$ to the atmosphere. $CO_2$ is known to be
emitted at the reservoir surface by diffusion at the air-water interface and downstream of
dams or powerhouses by degassing and along the river course. In this study, we quantified
total $CO_2$ emissions from the Nam Theun 2 Reservoir in the Mekong River watershed. The
study started in May 2009, less than a year after flooding and just a few months after the
maximum level was first reached and lasted until end of 2013. We tested the hypothesis that
soils from the drawdown area would be a significant contributor to the total $CO_2$ emissions.
Total inorganic carbon, dissolved and particulate organic carbon and $CO_2$ concentrations were
measured in four rivers of the Nam Theun watershed at nine stations in the reservoir (vertical
profiles) and at 16 stations downstream of the monomictic reservoir on a weekly to monthly
basis. $CO_2$ bubbling was estimated during five field campaigns between 2009 and 2011 and
on a weekly monitoring, covering water depths ranging from 0.4 to 16m and various types of
flooded ecosystems in 2012-2013. Three field campaigns in 2010, 2011 and 2013 were
dedicated to the soils description in 21 plots and the quantification of soil $CO_2$ emissions from
the drawdown area. On this basis, we calculated total $CO_2$ emissions from the reservoir and
carbon inputs from the tributaries. We confirm the importance of the flooded stock of organic
matter as a source of C fuelling emissions and we show that the drawdown area contributes,
depending on the year, from 50% to 75% of total annual gross emissions in this flat and
shallow reservoir. This overlooked pathway in terms of gross emissions would require an in-
depth evaluation for the soil OM and vegetation dynamics to evaluate the actual contribution
of this area in terms of net modification of gas exchange in the footprint of the reservoir, and
how it could evolve in the future.
**1   Introduction**
Carbon dioxide ($CO_2$) emissions from inland waters were recently revisited and it appears that
emissions from freshwater reservoirs contribute significantly despite the disproportionally
small surface area of these systems (Barros et al., 2011;Raymond et al., 2013;Deemer et al.,
2016). The $CO_2$ production and subsequent emissions in reservoirs result from the




degradation of the flooded organic matter (OM) and the OM originating from the watershed
(Galy-Lacaux et al., 1997b;Abril et al., 2005;Guérin et al., 2008;Barros et al., 2011;Teodoru
et al., 2011). As the amount of labile OM originating from the flooded soils and biomass
decreases with time due to the progressive mineralisation of the carbon stock, emissions
decrease progressively with reservoirs ageing (Abril et al., 2005;Barros et al., 2011). $CO_2$
emissions are higher in tropical reservoirs than in temperate and boreal ones, a latitudinal
difference attributed to the enhancement of OM degradation with temperature (Barros et al.,
2011;Marotta et al., 2014;Yvon-Durocher et al., 2014). Emissions occur through diffusion at
the air-water interface of the reservoir and from rivers downstream of dams (Abril et al.,
2005;Guérin et al., 2006;Kemenes et al., 2011). At the surface of reservoirs, it is well known
that emissions vary significantly spatially and temporally. Spatial variations can be higher
than temporal variations (Roland et al., 2010;Teodoru et al., 2011;Zhao et al., 2013;Pacheco
et al., 2015). Thus, the integration of both temporal and spatial variations is mandatory for the
determination of accurate emission factors.
Recently, the importance of the drawdown emissions was pointed out as a significant source
of $CH_4$ in the Three Gorges Dam (Chen et al., 2009;Chen et al., 2011;Yang et al., 2012) and a
very minor source at Nam Theun 2 Reservoir (NT2R) (Serça et al., 2016). $CO_2$ emission from
the drawdown area was only measured in agricultural plots of the drawdown area of the Three
Gorges Dam (Li et al., 2016). However, the hypothesis of significant $CO_2$ emissions from
those soils seasonally flooded and exposed to air was never tested in unmanaged drawdown
area representative of tropical reservoirs with large water level variations. In the present
study, we measured $CO_2$, organic and inorganic carbon concentrations and physico-chemical
parameters at 9 stations in the NT2R and 16 stations downstream of the dam and the
powerhouse. This weekly to fortnightly sampling was conducted in order to estimate
emissions from the reservoir surface and downstream emissions during 4.5 years of
monitoring after impoundment. We also measured $CO_2$ emissions from the large drawdown
area of the NT2R that represented seasonally up to 65% of the maximum reservoir area
during the study. The spatial, seasonal and interannual variation of emissions by all the above-
listed pathways and their contribution to total gross $CO_2$ emissions will be discussed.



## 2    Material and Methods

### 2.1    Study site

The NT2R is located in Lao People Democratic Republic's (Lao PDR), it was impounded in April 2008 and was commissioned in April 2010. It floods 489 km$^2$ of very diverse types of ecosystems including forest, agricultural soils and wetlands (Descloux et al., 2011). Geological formations responsible for the soil development in the NT2R area are mainly composed by more or less consolidated sedimentary rocks (Lovatt Smith et al., 1996;Smith and Stokes, 1997). The parental rocks belong to the Khorat group and Phon Hong group formations (Cretaceous) with two main lithologies: (1) late cretaceous Maha Sarakham formation (i.e., evaporites and mudstones) and (2) aptian Khot Kruat formation (i.e., mainly fluvial formation of red siltstones and sandstones)

The NT2R, described in details in Descloux et al. (2016);Deshmukh et al. (2016);Guérin et al. (2016) is under the influence of a monsoon subtropical climate with three main seasons: the cold dry season (CD, from mid-Oct. to mid-Feb.), the warm dry season (WD, from mid-Feb. to mid-June) and the warm wet season (WW, from mid-June to mid-Oct.). Owing to the large seasonal variations of the river discharges in the region, the reservoir area decreased down to 170 km$^2$ in the 2011 WD season during the course of the study. On the opposite, the surface of the drawdown area reached its maximum (320 km$^2$) when the water level was the lowest. During the monitoring, the wettest years were 2011 and 2013 with an average water discharge in the reservoir of ~270 m$^3$ s$^{-2}$ whereas the driest year was 2012 with a discharge 230 m$^3$ s$^{-2}$. In 2011, in this single year the reservoir had the largest water level variations with the largest surface area of the monitoring in the wet season (491 km$^2$) and the smallest of the monitoring in the WD season (168 km$^2$). The NT2R is a trans-basin reservoir with two downstream sections: one below the powerhouse and one below the Nakai Dam (Figure 1). Except during the occasional use of the spillways, only 2m$^3$ s$^{-1}$ of water are discharged downstream of the Nakai Dam in the Nam Theun River and around 240 m$^3$ s$^{-1}$ are released to the powerhouse, the regulating pond and finally the artificial downstream channel before water reaches the Xe Bangfai River (Figure 1).



## 2.2 Sampling strategy

The $CO_2$ and $O_2$ concentrations in water and the water temperature were determined in
surface waters of six pristine rivers and three rivers under the influence of the reservoir (10
stations) and in the artificial channel (5 stations) whereas it was done along vertical profiles in
the reservoir (9 stations) and the regulation pond (1 station) (Figure 1). At all sites located
downstream of the powerhouse, sampling was done weekly (from March 2010 to December
2013) whereas it was done fortnightly in incoming pristine rivers and in the reservoir (from
May 2009 to December 2013). The stations RES1-RES3 flooded dense forest, the stations
RES4-RES6 flooded degraded forest, the station RES7 flood swamps and the station RES8
flooded a rice field area (Descloux et al., 2011;Guérin et al., 2016). The station RES9 is
located at the water intake, an area of continuous vertical mixing of the water column, where
$CH_4$ emissions are enhanced (Guérin et al., 2016). Degassing of $CO_2$ was calculated below
the Nakai Dam, just below the turbines at TRC1, below the regulating dam (RD on Figure 1)
and at the aeration weir (AW on Figure 1). Bubbling of $CO_2$ was determined during five field
campaigns covering different seasons and sites in 2009, 2010 and 2011, and during a weekly
monitoring from March 2012 to August 2013 at seven stations. In the drawdown area, soil
description was conducted in June 2010 at six sites and $CO_2$ emissions were repeatedly
measured at 21 plots over those sites in June 2010, 2011 and 2013.

## 2.3 In situ measurements and water analysis

Vertical profiles of $O_2$, pH and temperature were measured in situ at all sampling stations
with a multi-parameter probe Quanta® (Hydrolab, Austin, Texas) since January 2009. In the
reservoir, the vertical resolution was 0.5 m down to 5 m and and 1 m deeper. Surface and
deep-water samples for $CO_2$, dissolved organic carbon (DOC), particulate organic carbon
(POC) and dissolved inorganic carbon (DIC) concentrations were taken with a surface water
sampler (Abril et al., 2007) and a UWITEC sampling bottle, respectively. Water samples for
$CO_2$ determination were stored in serum glass vials, capped with butyl stoppers, sealed with
aluminium crimps and preserved (Guérin and Abril, 2007). $CO_2$ concentrations were
determined by the headspace technique and using the solubility coefficient of Weiss (1974) as
in Guérin et al. (2006). The $CO_2$ partial pressure in headspace was determined by gas
chromatography (GC) (SRI 8610C gas chromatograph, Torrance, CA, USA) equipped with a
flame ionization detector and a methanizer (Chanudet et al., 2011). Commercial gas standards
(400, 1000 and 3000 ppmv, Air Liquid "crystal" standards) were injected after every 10



samples for calibration. Detection limit was < 1 ppmv in headspace and duplicate injection of
samples showed reproducibility better than 5%. For TIC, DOC and POC, analyses were
performed with a Shimadzu TOC-$V_{CSH}$ analyser. Filtered (0.45 μm, Nylon) and unfiltered
samples were analysed for TIC and TOC. POC was calculated by the difference between
TOC and DOC concentrations in unfiltered and filtered samples. The detection limit was 8
μmol $L^{-1}$ and uncertainty was 2.0 μmol $L^{-1}$ on TOC and DOC and 2.8 μmol $L^{-1}$ on POC.
**2.4   Organic and inorganic carbon inputs from the watershed to the reservoir**
Carbon inputs were calculated on a monthly basis using monthly average of the river
discharge of the four main tributaries of the NT2R. The Nam Theun River contributed 32% of
the total discharge while Nam Xot, Nam On and Nam Noy (not monitored for
biogeochemistry) contributed 24%, 23 and 22% respectively. For the Nam On River, the
specific water discharge and POC, DOC, TIC and $CO_2$ from this river were used. For the
other rivers, the specific water discharge of each river was used together with the average
DOC, POC, TIC and $CO_2$ from Nam Theun, Nam Phao and Nam Xot Rivers all located in the
Nam Theun watershed. Note that the Nam Phao reaches the Nam Theun River downstream of
the Nakai Dam but we used this dataset together with the ones from other rivers to calculate
the carbon inputs since the physico-chemical parameters and carbon concentrations are not
different from other rivers in the watershed.
**2.5   Diffusive fluxes and degassing**
Diffusive fluxes at the air-water interface of the reservoir were calculated from the surface
$CO_2$ concentrations, wind speed and rainfall rates using the gas transfer velocity formulations
of Guérin et al. (2007) and MacIntyre et al. (2010) as already described for $CH_4$ fluxes from
this reservoir (Deshmukh et al., 2014;Guérin et al., 2016). Based on physical modelling and in
situ measurements (Chanudet et al., 2012), we determined that the station RES9 located at the
water intake is representative of an area of about 3 $km^2$ (i.e. 0.6 % of the reservoir water
surface at full reservoir water supply), whatever the season (Guérin et al., 2016). This area
was therefore used to extrapolate specific diffusive fluxes from this station. For other stations,
diffusive fluxes are calculated with the daily meteorological parameters and reservoir water
surface area from the capacity curve. Degassing downstream of the powerhouse, the
regulating dam and the aeration weir, all located along the artificial channel and downstream
of the Nakai Dam (Figure 1), were computed using the $CO_2$ concentration upstream and

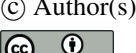



downstream of these civil structures and the water discharge as in Deshmukh et al. (2016) for
$CH_4$. The diffusion from the rivers and artificial channel below the powerhouse and the dam
was calculated using a constant gas transfer velocity of 10 cm h$^{-1}$ (Deshmukh et al., 2016).

**2.6  CO$_2$ bubbling**

Bubbling of $CO_2$ was determined with funnels (Deshmukh et al., 2014) during five field
campaigns covering different seasons (between May 2009 and June 2011), and during a
weekly monitoring from March 2012 to August 2013. During this monitoring, spatial
variation was explored through measurements spread over six stations (Fig. 1) representative
of the different types of flooded ecosystems (dense and medium forests, light and degraded
forest and agricultural lands as determined by Descloux et al. (2011)), and with different
depths (from 0.4 to 16 m) at each station. We stopped measuring bubbling at sites deeper than
16m after no ebullition was observed during the first three campaigns. Bubble samples were
taken with a 50 mL-syringe and the syringe was immediately connected to a N$_2$-preflushed
10-mL serum vial, leading to a dilution factor of 5/6 (Guérin et al., 2007). Gas samples were
analysed with the GC described above.

**2.7  Soil descriptions and CO$_2$ fluxes from the drawdown area**

Since the drawdown area of the NT2R could represent up to 65% of the surface area of the
reservoir at the end of the WD season, emissions from this major area under the influence of
flooding were evaluated. Soils types were determined together with $CO_2$ emissions. Soil
description was carried out in June 2010 at 6 sites and soils from the station RES4S plot were
characterized in details in June 2013 (Figure 1, Table 1). Four sites were selected in the Nam
Theun River riparian's area (NMR, RES2S, RES4S, RES8S'), one site in the flooded primary
forest (RES3S) and one site in the flooded agricultural area (RES8S). Soil study was
conducted through soil catenae of 2 to 4 soils profiles from the pristine soils on top ("upland"
samples) to the shoreline of the reservoir ("shoreline" samples). One or two other soils
profiles were performed in between ("interm.up" and "interm.down" samples). Soil sampling
was performed with an Edelman soil corer down to a depth of 1m, but only 0-20cm depth
samples were considered in this study. Information on horizon depth, soil texture and
structure (e.g., compactness, porosity), color (Munssel chart for soil color), soil fauna activity
and pedological features (e.g., redoximorphic features, concretions) were provided through
soil description in the field. Samples for C, N, and pH were selected following the horizons



apparition for each soil profile. They were manually decompacted and stored in plastic bags.
Back in the laboratory, soil samples were dried out at room temperature under a laminar flow
hood, sieved at 2 mm and properly split in two representative subsamples. One of the
subsample was crushed with an agate mortar before chemical analysis. The non-crushed
subsample was dedicated to soil pH and granulometric measurements. C and N analysis
where performed with a Elementar Vario EL III C/N/S analyser and soil pH measurements
were performed in ultrapure water (18.2 MΩ) following ISO 11464 standard procedure.
At the 6 sites, fluxes were measured along the soil moisture gradient between the shoreline
and the zone not impacted by the reservoir water level fluctuation. Three to four sites with
contrasting moisture content were selected at each site. At those six sites, fluxes were
measured at 21 plots in total and 40 $CO_2$ fluxes were gathered, mostly in duplicates (from 1 to
4 replicates) (Table 2). $CO_2$ emissions were measured during 3 field campaigns in 2010, 2011
and 2013 using stainless steel chamber (volume 12 L, 0.08 m$^2$) described in Serça et al.
(1994) and Serça et al. (2016). At each site, two chambers were deployed in parallel and the
collars were installed at least 1 hour prior to measurement. Air samples were taken and stored
with the same methodology as for bubbling samples every 15 minutes in each chamber before
subsequent GC analysis. It has to be noted that soil studies and measurement of fluxes were
restricted for safety reason due to the high density of unexploded ordnances (UXO) from the
sixties and seventies in that area.
**3    Results**
**3.1    Temperature, oxygen, organic and inorganic carbon in the Nam Theun**

231          **watershed and carbon inputs to the reservoir**

In the rivers of the Nam Theun watershed, the water temperature was 24.5±0.2°C ranging
from 13.5 to 32.0°C and pH was 6.83±0.03 (4.75-8.95, n=405). The Nam On River was, on
average, less oxygenated (77±2%) than the others. It is characterized by the highest DOC
concentrations (222±11 μmol L$^{-1}$, n=93), and amongst the highest $CO_2$ concentrations (59±6
μmol L$^{-1}$, n=107) and the lowest TIC concentration (237±11μmol L$^{-1}$, n=107) (Figure 2). The
Nam Phao and the Nam Theun Rivers are not significantly different in terms of POC, DOC,
TIC and $CO_2$ concentrations (Figure 2). During the monitoring, the average DOC in the Nam
Phao was 87±4 μmol L$^{-1}$ (n=82) and 108±4 μmol L$^{-1}$ (n=97) in the Nam Theun, that is more
than two times lower than in the Nam On. TIC was 40% higher in the Nam Theun and Nam

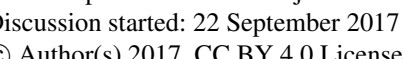



Phao Rivers than in the Nam On (Nam Phao: $380\pm12$ μmol $L^{-1}$, n=82; Nam Theun: $379\pm15$
μmol $L^{-1}$, n=97) (Figure 2). $CO_2$ in the Nam Theun River ($54\pm5$ μmol $L^{-1}$, n=105) and in the
Nam Phao ($46\pm5$ μmol $L^{-1}$, n=86) contributed around 15% of TIC whereas it was almost 25%
in the Nam On. The Nam Xot River had amongst the lowest DOC ($90\pm3$ μmol $L^{-1}$, n=93),
TIC ($272\pm12$ μmol $L^{-1}$, n=94) and $CO_2$ ($45\pm3$ μmol $L^{-1}$, n=110) concentrations (Figure 2).
Comparing results from all rivers, we could not find any significant differences in POC
concentration. In all rivers during this monitoring, the average POC was $28\pm2$ μmol $L^{-1}$
(n=200) and contributed less than 20% of the total organic carbon (DOC+POC) in this
watershed (Figure 2). We could not identify any clear seasonal pattern for POC, DOC TIC
and $CO_2$ concentrations in the four rivers of the Nam Theun watershed (Figure 2).
As reported in Descloux et al. (2016), the average total water discharge in the reservoir is 238
$m^3$ $s^{-1}$ ranging from 6 $m^3$ $s^{-1}$ during the WD seasons to 2061 $m^3$ $s^{-1}$ during the WW seasons.
Carbon input to the reservoir as DOC, POC and TIC ranged from $32.2\pm1.3$ GgC $yr^{-1}$ in 2010
to $46.2\pm1.5$ GgC $yr^{-1}$ in the wet year 2011 (Figure 3). During the monitoring, TIC represented
60 to 70% of the carbon inputs to the reservoir (Figure 3).

## 3.2 Vertical profiles of temperature, $O_2$, $CO_2$ and organic carbon in the reservoir water column

At the stations RES1-RES8, the typical vertical distributions of temperature, $O_2$, DOC, POC
and $CO_2$ for the three seasons at various sampling stations are shown in Figure 4. As already
described in details in Guérin et al. (2016), during the four years of monitoring, the reservoir
water column was thermally stratified during the warm seasons with thermocline at $4.5\pm2.6$
and $5.8\pm4.8$ m depths during the WD and WW seasons, respectively. As a consequence of
thermal stratification, the warm epilimnic waters are well oxygenated (>80% saturation)
whereas the hypolimnion is anoxic (Figure 4). Occasionally, sporadic and local
destratification occurred during high water inflow in the WW season. During the CD season,
temperature and $O_2$ decreased gradually with depth or $O_2$ concentration was constant from the
surface to the bottom of the water column (Figure 4). After the power plant commissioning,
the water column located near the Turbine Intake (RES9) got totally mixed as revealed by the
homogeneous temperature and $O_2$ profiles from the surface to the bottom (Figure 4). pH
always decreased from the surface to the bottom with, on average during the monitoring,



surface pH = 6.66±0.02 (5.21-8.76, n=1316) and hypolimnic pH = 6.15±0.01 (4.88-8.00,
n=1488).
Over the monitoring period at the stations RES1-RES8, the average $CO_2$ concentration in the
water column was 389±9 µmol $L^{-1}$ and ranged from 0.3 to 4770 µmol $L^{-1}$ (n=3698). It
decreased from 544±24 µmol $L^{-1}$ in 2010 to 154±9 µmol $L^{-1}$ in 2013. During the WD and
WW seasons, $CO_2$ concentrations increased with water depth and often showed a maximum
gradient at or just below the thermocline (Figure 4). For the years 2010 to 2013, the average
$CO_2$ concentrations in the water column during the WD and WW seasons were always 50%
higher than in the CD season (Figure 4). DOC concentrations averaged 181±1 µmol $L^{-1}$ and
ranged from 12.5 to 569 µmol $L^{-1}$ (n=3068). For the years 2010, 2011 and 2012 we observed
a significant decrease of DOC in the water column from year to year with average DOC
concentrations 208±3 µmol $L^{-1}$ in 2010, 190±3 µmol $L^{-1}$ in 2011 and 177±2 µmol $L^{-1}$ in 2012.
In 2013, the DOC was not significantly lower than in 2012 (175±2 µmol $L^{-1}$). From 2010 to
2013, DOC concentrations were about 25% higher in the WD and WW seasons than in the
CD season. Whatever the year, the average epilimnic DOC concentration was 30% higher
than in hypolimnic water. POC concentration was 63±2 µmol $L^{-1}$ (n = 2488). POC in
hypolimnic waters (92±3 µmol $L^{-1}$) was almost twice higher than in the epilimnion (45±2
µmol $L^{-1}$) (p < 0.0001, t-test). The POC in the epilimnion decreased significantly from 41±4
µmol $L^{-1}$ in 2010 to 23±2 µmol $L^{-1}$ in 2013 in the epilimnion (p < 0.0001). POC in
hypolimnic waters did not show any consistent trend with yearly average values being 87±6
µmol $L^{-1}$ in 2010, 67±6 µmol $L^{-1}$ in 2011, 104±7 µmol $L^{-1}$ in the wet year 2012 and 83±5
µmol $L^{-1}$ in 2013. No clear seasonal variation was observed.
At the station RES9 where the presence of the water intake enhances vertical mixing of the
water column leading to the transport of bottom water to the surface, the water column is not
thermally stratified and always oxygenated from the surface to the bottom after the reservoir
was commissioned in April 2010 (Guérin et al., 2016) (Figure 4). Since commissioning, $O_2$
saturation was 60±2 % over the water column. The water column was significantly more
oxygenated during the overturn in the CD (74±3%) than in the WW and WD season (56±2%)
(p < 0.0001, t-test) and significantly more oxygenated (p < 0.0001) in the wet year 2011
(70±3%) than in 2010 and 2012 (56±3%). In 2013, which was an average hydrological year,
the the water column was well oxygenated with 71±1% suggesting of improvement of the





water quality. $CO_2$ concentrations were almost constant from the surface to the bottom and
averaged 216±13 µmol $L^{-1}$ over the whole monitoring period (n = 512) (Fig. 4). $CO_2$
concentration in the water column decreased from 311±32 µmol $L^{-1}$ in 2010 down to 28±2
µmol $L^{-1}$ in 2013. Concentrations in the WW and WD seasons were similar 204±14 µmol $L^{-1}$
and more than two times higher than during the CD season (105±6 µmol $L^{-1}$). POC
concentration was 25±1 µmol $L^{-1}$ (n=431) and DOC was 157±2 µmol $L^{-1}$ (n=642) over the
whole water column and both follow the same seasonal variations and temporal variations as
described for the other stations.

### 310   3.3   Reservoir surface $CO_2$ concentration and diffusive fluxes

The reservoir surface $CO_2$ concentrations (n=1067) ranged from 0.3 to 970 µmol $L^{-1}$ (Figure
5a,b) and diffusive fluxes ranged from -40.4 up to 2694.9 mmol $m^{-2}$ $d^{-1}$ (Figure 5c,d). Most of
the dataset (85% of all measurements) showed $CO_2$ supersaturation with respect to the
atmosphere. In 2009 (from May to December), surface concentrations and diffusive fluxes
from all nine sampling stations located in the reservoir were statistically similar (p > 0.05,
ANOVA test). The average surface concentration was 68.2±47.9 µmol $L^{-1}$ and the diffusive
flux was 101.6±137.7 mmol $m^{-2}$ $d^{-1}$.
From 2010 to 2013 at the stations RES1 to RES8, the yearly average surface $CO_2$
concentrations decreased significantly from 62.7±3.6 to 32.7±3.2 µmol $L^{-1}$ while diffusive
fluxes decreased from 89.8±10 to 13.7±4.7 mmol $m^{-2}$ $d^{-1}$ without any significant spatial
variations (p > 0.05, ANOVA test). Over the 2010-2012 period, the highest concentration and
fluxes were always observed in the WD season (70±3 µmol $L^{-1}$ and 90±9 mmol $m^{-2}$ $d^{-1}$), they
decreased down to 51±3 µmol $L^{-1}$ and 65±8 mmol $m^{-2}$ $d^{-1}$ in the WW and reached their
minima in the CD season (45±3 µmol $L^{-1}$ and 22±2 mmol $m^{-2}$ $d^{-1}$) (Figure 5 a,c). In 2013, the
reservoir was a net CO2 sink from March to August (-11±2 mmol $m^{-2}$ $d^{-1}$, n=96) and
emissions in the CD season was 66±9 mmol $m^{-2}$ $d^{-1}$ (n=41) that is three times higher than
usually observed for that season.
At the water intake (RES9) after the commissioning, surface concentrations and diffusive
fluxes were statistically different from the other stations and were significantly higher as
already observed for $CH_4$ (Guérin et al., 2016). The average surface $CO_2$ concentrations at
RES9 were 287±350 and 184±234 µmol $L^{-1}$ for the year 2010 and 2011, respectively that is
three-fivefold higher than the average at the other stations (Figure 5b). In 2012, surface $CO_2$




concentrations at RES9 dropped down to 65±23 µmol L$^{-1}$, still almost twice the concentration
at the other stations. In 2013, surface concentration at RES9 was not statistically different
than at the other station in the reservoir (33±4 µmol L$^{-1}$ in 2013). On an annual basis, the
diffusive fluxes at RES9 decreased from an average of 745±195 to 18±9 mmol m$^{-2}$ d$^{-1}$
between 2010 and 2013 (Figure 5d). The same seasonality as described before was observed
at RES9 with a exacerbated effect at the transition between the WD and WW seasons since
diffusive fluxes were then up to 17-fold higher than the average fluxes at the other stations for
that same period (Figure 5c,d).
Monthly emissions by diffusive fluxes varied by two orders of magnitude between 2009 and
2012. Superimposed to the general decrease of emissions with time, we observed very
significant seasonal variations with emissions peaking during the transition between the WD
and WW seasons, even though the reservoir water surface was at its minimum (Figure 5e).
The annual diffusive $CO_2$ emission from the reservoir was 730.0±46.2 Gg($CO_2$) year$^{-1}$ in 2009
and dropped down by a factor of six in 2013 (118±11.5 Gg($CO_2$) year$^{-1}$) (Figure 5f).

### 3.4 $O_2$, organic carbon and $CO_2$ downstream of the reservoir

After the commissioning, immediately downstream of the power station (station TRC1), the
average $O_2$ concentration was 174±58 µmol L$^{-1}$, that is, 67±20% saturation (n=189) and pH
was 6.55±0.04 (n=234). Further downstream, the $O_2$ concentration always increased and the
$O_2$ saturation downstream of station DCH4 located 30 km from the turbines was always
around 100% saturation in the artificial downstream channel (average 100.4%, n=146). Just
below the regulating dam, in the Nam Kathang River (NKT3), the average $O_2$ concentration
was 237 µmol L$^{-1}$, that is, 93% saturation (n=120). There was no marked interannual change
in the $O_2$ concentration. At DCH4, pH increased to 7.17±0.04 (n=186).
On average at all the stations in between TRC1 and DCH4, DOC concentration was 159±2
µmol L$^{-1}$ (n=1366) over all stations for all years between 2009 and 2013. DOC decreased
from 187±2 µmol L$^{-1}$ in 2010 (n=272) to 157±2 µmol L$^{-1}$ in 2013 (n=303). Average POC was
25±1 µmol L$^{-1}$ (n=818) for all years between 2009 and 2013, and followed interannual
variations already observed for the reservoir, i.e. higher POC concentration in the WW season
of 2012.




$CO_2$ concentration below the Powerhouse (TRC1), which receives water from the station
RES9 in the reservoir after the water transiting through the turbines, varied by almost three
orders of magnitude; ranging from 1.4 to 856 µmol $L^{-1}$ with an average of 153±14 µmol $L^{-1}$ (n
=199). The $CO_2$ concentrations varied seasonally with maximum concentrations at the end of
the WD season, and minimum at the end of the CD season. Below the powerhouse, along the
longitudinal transects from TRC1 to DCH4, surface $CO_2$ concentration decreased by a factor
of three within a distance of 30 km during the WD and WW seasons (from 267±34 to 90±10
µmol $L^{-1}$ and from 235±28 to 70±8 µmol $L^{-1}$ respectively for WD and WW). In the CD
season when $CO_2$ concentrations were lower, the decrease in concentration with distance from
the dam was only by a factor of two (from 49±8 to 30±4 µmol $L^{-1}$). Between 2010 and 2013
for all stations in the downstream channel (TRC1 to DCH4), annual average $CO_2$
concentrations decreased by a factor of 7 from 182±9 µmol $L^{-1}$ to 24±2 µmol $L^{-1}$. On average,
$CO_2$ concentration reached down to 56±5 µmol $L^{-1}$ at DCH4 which is in the same order of
magnitude as the concentrations found in the pristine Xe Bangfai River (XBF1, 60±2 µmol $L^{-1}$
$^1$, n=64), Nam Kathang Noy River (NKT1, 35±3 µmol $L^{-1}$, n=47) and Nam Kathang Gnai
River (NKT2, 82±10 µmol $L^{-1}$, n=70).
Immediately downstream of the Nakai Dam (NTH3) after the commissioning, the average $O_2$
concentration was 224 µmol $L^{-1}$, that is 87% saturation (n=73), and the concentration
increased further downstream. pH was 6.84±0.06 (n=166). Average DOC concentration was
166±2 µmol $L^{-1}$ (n=653) and decreased from 197±4 µmol $L^{-1}$ in 2010 (n=147) to 162±3 µmol
$L^{-1}$ (n=127) in 2013. The average POC concentration was 50±5 µmol $L^{-1}$ (n=7) and $CO_2$
concentration was 67±9 µmol $L^{-1}$ (n=77). The $CO_2$ concentration decreased by a factor of two
(40±5 µmol $L^{-1}$, n=54) within the next 10 km below the dam (down to NTH4, Figure 1) where
pH was 7.19±0.06 (n=97). At NTH4, the observed concentrations were in the same order of
magnitude than the concentrations in the pristine rivers in the same watershed (53±6 µmol $L^{-1}$
at NPH1 in the Nam Phao River, n=59).
**3.5  $CO_2$ emissions downstream of the reservoir**
After the commissioning, the annual average diffusive fluxes downstream of the powerhouse
decreased from 482±603 mmol $m^{-2}$ $d^{-1}$ in the year 2010 (-32-33762 mmol $m^{-2}$ $d^{-1}$) to 32±8
mmol $m^{-2}$ $d^{-1}$ (-39-216 mmol $m^{-2}$ $d^{-1}$) in 2013 (not show). They followed the same seasonal
dynamics as the $CO_2$ concentrations and they decrease with the distance from the



powerhouse. Total emissions by diffusion from the downstream channel decreased from
$14\pm12$ $Gg(CO_2)$ year$^{-1}$ in 2010 to $1.3\pm0.5$ $Gg(CO_2)$ year$^{-1}$ in 2013 (Figure 6a). Degassing in
the whole downstream channel (including degassing below the turbines, the regulating pond
and the aeration weir) reached up to 28.5 $Gg(CO_2)$ month$^{-1}$ just after the commissioning of the
reservoir when the water was released for the first time (Figure 6a). During the monitoring,
60-90% of the annual degassing occurred within 3-4 months of transition between the WD
and WW seasons corresponding to the seasons when the hypolimnic waters were the most
enriched in $CO_2$ (Figure 6a). Total degassing decreased from $80\pm36$ $Gg(CO_2)$ year$^{-1}$ in 2010
to $8\pm4$ $Gg(CO_2)$ year$^{-1}$ in 2013 (Figure 6b).
Disregarding periods of spillway releases from April to June 2009 for water level regulation
and in September-October 2011 during the flood, degassing downstream of the Nakai Dam
(up to 0.48 $Gg(CO_2)$ month$^{-1}$) is usually 10 times lower than degassing in the downstream
channel because of (1) the low continuous water discharge at the Nakai Dam (2 m$^3$ s$^{-1}$) and
(2) the withdrawal of the water from the reservoir epilimnion (Deshmukh et al., 2016) (Figure
6a). However, during the use of spillways for water level regulation in the reservoir,
degassing reached up to 26 $Gg(CO_2)$ month$^{-1}$ in 2009 before the commissioning and 4 to 10
$Gg(CO_2)$ month$^{-1}$ during the occasional uses in October 2010 and September 2011 (Figure
6a). As determined from the longitudinal profiles of $CO_2$ concentrations downstream of the
dam, diffusive emissions from the Nam Theun River that are actually attributable to the
NT2R occurred within the first 10 km below the dam as it was also the case for $CH_4$
(Deshmukh et al., 2016). The annual average diffusive $CO_2$ fluxes were $126\pm137$ and
$288\pm346$ mmol m$^{-2}$ d$^{-1}$ in 2010 and 2011 respectively. The annual average diffusive $CO_2$ flux
was one order of magnitude lower in 2013 ($24\pm68$ mmol m$^{-2}$ d$^{-1}$) (not show). The total
emissions by diffusion and degassing resulting from these fluxes ranged between $5.5\pm0.1$
$Gg(CO_2)$ year$^{-1}$ in 2010 and $0.14\pm0.06$ $Gg(CO_2)$ year$^{-1}$ in 2013 (Figure 6b).
On a yearly basis, emissions downstream of NT2R decreased from $99.7\pm25.3$ to $15.0\pm6.5$
$Gg(CO_2)$ y$^{-1}$ between 2010 and 2013 (Figure 6d). Before the reservoir commissioning in
2009, emissions were dominated by degassing due to spillway releases. After the
commissioning, emissions were dominated by degassing in the downstream channel which
contributed 80-90% of total downstream emissions.



### 3.6 $CO_2$ bubbling


The $CO_2$ content in the sampled bubbles was 0.29±0.37% (n=2334) and no bubbles was ever
observed for depth higher than 16 m. On average, the $CO_2$ bubbling was 0.16 ± 0.24 mmol m$^{-2}$
$^2$ d$^{-1}$ (0-2.8 mmol m$^{-2}$ d$^{-1}$). Considering the water surface variations, the monthly ebullitive
$CO_2$ emissions ranged from 0.04±0.06 to 0.11±0.16 Gg($CO_2$) month$^{-1}$. $CO_2$ bubbling was
constant around 1.1±2.2 Gg($CO_2$) y$^{-1}$ throughout the monitoring.

### 3.7 $CO_2$ emissions from the drawdown area


Four types of pristine soils were identified in the six different studied catenae. Acrisols were
the most represented soils and were found at three sites (RES4S, RES8S and RES8'S) (Table
1). In the area with dense forest, soils were characterized as plinthosol (RES3S) and plinthic
ferralsol (RES2S) and the pedological cover at MNR site belong to planosol type soil (Table
1). At all sites, from upland pristine soils to the shoreline, stagnic properties were more and
more pronounced. Average organic carbon content (%C), nitrogen (%N) and C:N ratio were
1.84±0.10%, 0.14±0.01% and 12.83±0.30, in surface horizons, respectively. For those three
parameters, no statistical differences were obtained according to soil type, topography or
measurement site. Diffusive $CO_2$ fluxes ranged between 34±7 and 699±59 mmol m$^{-2}$ d$^{-1}$
(Table 2). The fluxes were not significantly correlated with the surface moisture ranging from
17.5 to 51.2% and temperature ranging from 18.1 to 34.2°C (Table 2). The fluxes neither
varied significantly with soil types, topography, measurement sites, nitrogen content or C:N
ratio (p > 0.05, ANOVA test). However, average fluxes at each site were significantly
correlated with the average C content (p=0.452). Without significant spatial variations related
to topography, humidity or temperature, we further consider the average of all fluxes that is
279±27 mmol m$^{-2}$ d$^{-1}$.
After the commissioning of the reservoir, emissions varied by three orders of magnitude.
Since a constant $CO_2$ emission is accounted for, the seasonal pattern of $CO_2$ emission from
the drawdown mimics the variation of the surface of that area (Figure 7). Monthly $CO_2$
emissions could reach up to 110.8±10.7 Gg($CO_2$) month$^{-1}$ by the end of the WD season when
drawdown area reaches its maximum whereas it decreased down to 0.6±0.1 Gg($CO_2$) month$^{-1}$
at the end of WW season when drawdown area reaches its minimum (Figure 7). Around 80-
90% of the annual emissions occurred within 4-6 months of transition period between the WD
and WW seasons (Figure 7). The lowest emissions from the drawdown area occurred during





454 the wet year 2011 ($386\pm16$ Gg($CO_2$) year$^{-1}$) and the highest emissions during the dry year

455 2012 ($572\pm20$ Gg($CO_2$) year$^{-1}$).On average from 2009 to 2013, emissions from the drawdown

456 area was $431\pm42$ Gg($CO_2$) year$^{-1}$.

457 **4 Discussion**

458 **4.1 $CO_2$ dynamic in the NT2R water column and downstream rivers**

459 The dynamics of $CO_2$ in the NT2R is highly dependent on the hydrology and hydrodynamics

460 of the reservoir as it has already been described for $CH_4$ (Guérin et al., 2016). During the

461 warm seasons (WD and WW) when the water column is thermally stratified, the vertical

462 profiles of $CO_2$ concentration in the water column are similar to profiles obtained in other

463 tropical or subtropical reservoirs (Abril et al., 2005;Guérin et al., 2006;Kemenes et al.,

464 2011;Chanudet et al., 2011) but also boreal reservoirs (Demarty et al., 2011). The high

465 concentrations measured in the hypolimnion suggest that the main source of $CO_2$ is located at

466 the bottom and very likely in the flooded soils, vegetation and sediments whereas the decrease

467 of $CO_2$ toward the surface suggest both consumption by primary production and/or loss to the

468 atmosphere (Galy-Lacaux et al., 1997b;St Louis et al., 2000;Abril et al., 2005;Guérin et al.,

469 2008;De Junet et al., 2009;Teodoru et al., 2011;Barros et al., 2011;Chanudet et al., 2011). In

470 the CD season, after the reservoir overturn, the average $CO_2$ concentration in the reservoir

471 water column decreases sharply (by 50% on average) and $CO_2$ concentration increases

472 regularly from the surface to the bottom of the water column. However, no $CO_2$ burst was

473 observed at the beginning of the CD season when the reservoir overturns. Therefore it is

474 reasonable to assume that the reservoir overturn has only a moderate impact on $CO_2$

475 emissions. This assumption is reinforce by the fact that during the same sampling, hot

476 moments of $CH_4$ emissions were captured (Guérin et al., 2016). As observed in most tropical

477 and subtropical reservoirs, the higher concentrations were observed during the warm seasons

478 (Abril et al., 2005;Kemenes et al., 2011;Chanudet et al., 2011) whereas the lowest were found

479 after reservoir overturn (Chanudet et al., 2011). A significant shift in the carbon

480 biogeochemical cycling occurred in the reservoir in 2013 with the reservoir water surface

481 becoming of $CO_2$ sink during the WD season and the beginning of the WW season (from

482 March to August). Although no major change was observed nutrient concentrations, the

483 number of phytoplanktonic cell was 50% higher in 2013 than 2012 (Unpublished, M Cottet

484 personal com.) indicating that primary production was significantly enhanced in 2013.

485 Despite the fact that the reservoir was a sink for the six months when $CO_2$ emissions are





usually the highest of the year, annual $CO_2$ emissions at the surface of the reservoir were only
50% lower in 2013 than in 2012. In 2013, $CO_2$ was mainly emitted in the CD after the period
of high biological productivity suggesting that the degradation of autochthonous OM fuel $CO_2$
emissions.
The maximum concentration and the highest $CO_2$ stock in the water column highly depend on
the age of the reservoir. In the NT2R, average $CO_2$ concentration was three times higher in
2010 than in 2013 and the maximum concentrations in 2010 was almost two times higher than
in 2013 (4771 μmol $L^{-1}$ in 2010 vs. 2649 μmol $L^{-1}$ in 2013). Those high concentrations are
slightly lower than the maximum concentration measured in the hypolimnion of the Petit Saut
Reservoir less than a year after it was flooded (Galy-Lacaux et al., 1997a;Abril et al., 2005).
Disregarding these high concentrations observed in the hypolimnion of the reservoir at the
end of the WD season and beginning of the WW season in 2009 and 2010, the $CO_2$
concentration in the NT2R are in the same range as concentrations in other older reservoir in
the tropics or the subtropics (Abril et al., 2005;Guérin et al., 2006;Chanudet et al.,
2011;Kemenes et al., 2011). This decrease during the first four years after impoundment is
very consistent with the decrease of the $CO_2$ concentration with the reservoir age as already
observed at the Petit Saut Reservoir (Abril et al., 2005), at the Eastmain I Reservoir (Teodoru
et al., 2012) or over multi-sites study (Barros et al., 2011).
Disregarding the station RES9 located at the water intake, no significant spatial variation of
$CO_2$ surface concentrations was found despite very significant differences of hypolimnic
concentration between stations located upstream of the Nakai Dam (RES1, 2 and 3) and
station located in areas close to the three main tributaries (RES6, 7 and 8). The average
hypolimnic concentrations at the stations RES1-3 were two times higher than at the stations
RES6-8. This difference is attributed to both (1) the difference in carbon density at the bottom
of the reservoir in those two contrasted areas in terms of submerged ecosystems (Descloux et
al., 2011) (see section 4.3) and (2) the difference in terms of water residence time between
those two zones (Guérin et al., 2016). Stations RES1-3 are located in areas with the longest
water residence time in the reservoir since the water mostly enters the reservoir in the RES6-8
area from the Nam Theun, Nam Noy and Nam On Rivers before being delivered to the water
intake (close to RES9) on the opposite side of NT2R which has a narrow and elongated shape
(Figure 1). Therefore, the water renewal in the RES6-9 area is high and $CO_2$ accumulates less



in the water column confirming the importance of the reservoir hydrology on the spatial
variability of dissolved gases in reservoirs (Pacheco et al., 2015;Guérin et al., 2016).
As found for $CH_4$, the main factor influencing the spatial variability of $CO_2$ in the water
column is the vertical mixing of the water column induced by the water intake located close to
RES9 (Deshmukh et al., 2016;Guérin et al., 2016). The design of the water intake enhances
horizontal water current velocities and vertical mixing which lead to the transport of bottom
waters to the surface. As a consequence, surface concentrations at RES9 were up to 30 times
higher than at other stations in 2010 and 2011 (Figure 5b). With the significant decrease of
concentrations in 2012 and 2013, the difference with other stations dropped to a factor of
four. These maximum surface concentrations at RES9 are up to 10 times higher than
concentrations found in other tropical reservoir in South America (Abril et al., 2005;Guérin et
al., 2006;Kemenes et al., 2011) and Lao PDR (Chanudet et al., 2011) showing that, as for
$CH_4$, $CO_2$ emissions can be enhanced upstream of water intake or dams.
Downstream of the reservoir in the Nam Theun River or the artificial channel, $CO_2$
concentrations follow the same seasonality as in the reservoir. Concentrations peak in June-
July at the transition between the WD and the WW season, and reach their minima in the CD
season. Downstream of the Nakai Dam, the concentrations are twice lower than downstream
of the powerhouse since mostly epilimnic water from the RES1 station is transferred
downstream of the dam. Within less than 10 km further downstream, concentrations are not
significantly higher than in pristine rivers of the watershed. Downstream of the powerhouse,
$CO_2$ concentrations in 2010 were in the same order of magnitude as in 10-20 years-old
reservoirs in South America flooding tropical forest (Abril et al., 2005;Guérin et al.,
2006;Kemenes et al., 2011) whereas four years after impoundment $CO_2$ concentrations were
two times lower than in 20-30 years-old reservoirs in Lao PDR (Chanudet et al., 2011). We
hypothesize that the low $CO_2$ concentration downstream of the NT2R result from a significant
degassing of the water at the water intake before the water is transferred downstream as
observed for $CH_4$ (Deshmukh et al., 2016;Guérin et al., 2016).

## 4.2   Total $CO_2$ emissions from the Nam Theun 2 Reservoir

From 2009 to 2013, total $CO_2$ emissions from NT2R show the same seasonal pattern (Figure
8a). The lowest total emissions occur in the CD season while the highest emissions occur at
the transition between the WD and the WW season when emissions by all individual



pathways reach their maximum. From 2010 to 2013, emissions at the transition between the
WD and the WW season between April and July contributed 47 to 61% of total emissions
suggesting that quantification of emissions based on two to four campaigns in a year might be
subject to caution since seasonality of emissions significantly affects emissions factors.
$CO_2$ bubbling follows the same seasonal variations, being triggered by water level and
concomitant hydrostatic decrease as for $CH_4$ (Chanton et al., 1989;Engle and Melack,
2000;Smith et al., 2000;Boles et al., 2001;Deshmukh et al., 2014) but its contribution is
negligible (<1%, Table 3). Low $CO_2$ emission by bubbling as also observed in temperate
reservoirs (Bevelhimer et al., 2016) is attributed to the higher solubility of $CO_2$ in water than
$CH_4$ which lead to the solubilisation of the majority of $CO_2$ as free $CO_2$ or as DIC.
The relative contribution of emissions downstream of the reservoir by degassing and diffusion
from rivers and channels at NT2R are low compare to most of the reservoirs that were studied
(Abril et al., 2005;Guérin et al., 2006;Kemenes et al., 2011;Bevelhimer et al., 2016) but the
contribution of this pathway is comparable to what was observed in boreal reservoirs (Roehm
and Tremblay, 2006) or in monomictic reservoirs from Lao PDR (Chanudet et al., 2011). The
downstream emissions contributed between 11% at the maximum in the wet 2011 year down
to 3% at the minimum in 2013 (Table 3 and Figure 8a). As for $CH_4$ at NT2R (Deshmukh et
al., 2016), the low downstream emissions are attributed to the significant degassing at the
water intake (station RES9) before the water reach the turbines and to the flush of $CO_2$ due to
the reservoir overturn in the CD season.
Emissions by diffusive fluxes at the surface of the reservoir increase by a factor of two by the
end of the WD season (Figure 5a) compare to the CD season from 2009 to 2012. The average
$CO_2$ emissions in 2009 and 2010 and in a lesser extend 2011 are in the same range as
emissions from the Petit Saut Reservoir during the first five years after impoundment (Abril et
al., 2005) and in the upper range of average $CO_2$ diffusive fluxes measured in older tropical
reservoirs (dos Santos et al., 2006;Kemenes et al., 2011;Yang et al., 2013) or in young boreal
reservoirs (Teodoru et al., 2011;Tadonléké et al., 2012). In 2012 and 2013, emissions from
NT2R by diffusive fluxes are still higher than most of the older Asian reservoirs (Wang et al.,
2011;Chanudet et al., 2011;Zhao et al., 2013;Xiao et al., 2013;Panneer Selvam et al., 2014)
and other Brazilian reservoirs flooding savannah (Roland et al., 2010;Pacheco et al., 2015).
The low emissions in the CD season from the first 3.5 years might mostly result from lower
heterotrophic activity due to the low temperature (down to 7°C in air in March 2011). The





high emissions in the CD season of 2013 as compared to CD season in 2011 and 2012 likely
originate from additional autochthonous OM. We hypothesise that the significantly higher
$CO_2$ emissions in the WD season result from the increase of the water residence time that
favour $CO_2$ accumulation in the water column (Abril et al., 2005) and the increase of
temperature that enhance aerobic and anaerobic degradation of OM and the production of $CO_2$
(Sobek et al., 2005). Although the reservoir area during the WD season is the smallest of the
year, emissions by diffusive fluxes are the highest (Figure 8a) highlighting the very
significant increase of $CO_2$ emissions from May to July every year, disregarding the year

588    2013.

This first estimation of the $CO_2$ emission from the drawdown area to the total emission from a
reservoir reveal that with a contribution ranging from 40 to more than 75%, it could be a
major $CO_2$ pathway to the atmosphere. These results from the NT2R cannot be generalized to
all reservoirs since its contribution is tightly link to the very high water level variations and
large surface area of the drawdown area (up to 320 km², Figure 7). However, areal fluxes
from the drawdown area are on average 2.5 times higher than the diffusive fluxes from the
reservoir water surface in 2009-2010 and six times higher than those fluxes in 2013 making
the soils in the area of influence of the reservoir a hotspot for $CO_2$ emissions compare to the
reservoir surface waters. The total emissions of reservoirs with contrasted hydrology
characterized by marked wet and dry seasons and large water level variations of 30% of the
total surface could have been significantly underestimated as it is the case for Petit Saut (~100
km²), Samuel (~280 km²), Balbina (~220 km²) or Three Gorges Reservoir (~400 km²) for
instance (Guérin et al., 2006;Kemenes et al., 2011;Li et al., 2016). This pathway is expected
to be more significant in flat bottom reservoirs than in valley type reservoirs in mountainous
regions and cannot be generalized on just the drawdown area without taking into account
hydrological water management and the local topography. At Petit Saut and NT2R at least, no
vegetation regrowth occurs in the drawdown areas. Soils at NT2R exhibit very clear
modification related to the flooding (stagnic features) confirming soil modification as also
observed in Canada (Furey et al., 2004) Australia (Watts, 2000) and France (Félix-Faure et
al., 2017). The ecosystems of the seasonally flooded area are therefore significantly modified
and $CO_2$ emissions from the drawdown must be accounted for in total gross emissions from
reservoirs. Although drawdown emissions cannot be neglected in terms of gross $CO_2$
exchange, the emissions resulting from the soil respiration are currently comparable to
pristine emissions (Table 2) and the impact of these area in terms of net emissions requires





further specific studies in these overlooked ecosystems. So far, we cannot predict future
evolution of $CO_2$ emissions in this area based on the available data. The consequence of the
flooding on the respiration rate of these soils may lead to a decrease of emissions with time or
a stabilization (see next section). Therefore, the net contribution of the drawdown zone to
emissions from the reservoir remains unclear and specifically requires research on soil OM
dynamics and would also require the inclusion of the vegetation dynamics when present.
This is the first comprehensive quantification of $CO_2$ emissions from a reservoir where all
known $CO_2$ pathways to the atmosphere were taking into account at one of the best spatial
and temporal resolution reported in the literature. We showed that downstream emissions and
emissions around the water intake are not negligible (~10% overall) and that the overlooked
drawdown area in $CO_2$ studies could be the main emission pathway of $CO_2$ to the atmosphere.
Overall, this study highlights that global estimate of $CO_2$ and $CH_4$ emissions from reservoir
are underestimated so far since relevant pathways like drawdown emissions in flat/shallow
reservoirs with large water level variations and downstream emissions in thermally stratified
reservoirs are missing in most site-specific studies used for extrapolations (Deemer et al.,
2016;Barros et al., 2011).

### 4.3 Source of organic matter fuelling the reservoir $CO_2$ emissions

In tropical reservoirs, the decrease of the $CO_2$ concentration in the water column and
subsequent emissions with the age of the reservoir (Figure 8b) is supposed to result from the
decrease of the aerobic and anaerobic mineralisation rate due to the exhaustion of labile OM
from the pool of soil and vegetation that was flooded during impoundment (Abril et al.,
2005;Guérin et al., 2008). In boreal reservoirs, the decrease of benthic $CO_2$ production is
sharp and after 3-5 years, most of the $CO_2$ production appears to be pelagic and is supposed
not to result from the flooded organic matter (Teodoru et al., 2011;Brothers et al., 2012). The
total $CO_2$ emissions were nine and three times higher than the carbon inputs from the
watershed to the NT2R in 2010 (32 GgC yr$^{-1}$) and 2013 (45 GgC yr$^{-1}$), respectively (Figure 3
and Table 3). It has to be noted that interannual variations of carbon inputs to the NT2R
(Figure 3) are not correlated with the regular decrease of total $CO_2$ emissions from year to
year (Figure 8b). It is therefore unlikely that most of $CO_2$ emissions result from the
mineralization of allochthonous OM but rather from the contribution of the flooded carbon
pool (soil and vegetation) which amount is decreasing with time. This is consistent with the



fact that at Petit Saut, even 10 years after flooding, the majority of the OM in the water
column has a terrestrial origin (De Junet et al., 2009). According to Abril et al. (2005) at Petit
Saut, total emissions (disregarding drawdown emissions which were not measured) were 9 to
6 times higher than carbon inputs from the watershed during the first 4 years for similar
carbon inputs which indicates a faster decrease of emissions in NT2R than at Petit Saut. This
sharp decrease of emissions at NT2R might be due to the fact that the flooded pool of OM and
therefore the amount of labile OM in NT2R was twice smaller than the amount of OM
flooded in the Petit Saut (Guérin et al., 2008;Descloux et al., 2011). We show here, as it was
done at Petit Saut (Guérin et al., 2008;Abril et al., 2005), that external sources of carbon are
not sufficient to fuel the $CO_2$ emissions from the NT2R and we attribute the decrease of
emissions with time to the exhaustion of the most labile fraction of the flooded pool of OM
which might be the main source of reactive carbon in the reservoir.
In the sub-tropical NT2R, $CO_2$ concentrations are always higher at the bottom than in the
epilimnic waters even during the CD season when the limited thermal stratification or its
absence do not favour hypolimnic $CO_2$ accumulation. The CD season is probably the most
favourable season to pelagic respiration as this process is enhanced by the re-oxygenation of
the water column (Bastviken et al., 2004). Since $CO_2$ concentration in the CD season is 50%
lower than in the warm seasons, we suggest that $CO_2$ is mostly produced in the sediment and
flooded soils and vegetation. Disregarding the station RES9 located at the water intake,
significant spatial variation of $CO_2$ hypolimnic concentrations were found between stations
located in the area of dense forest (RES1-3) versus stations located in areas close to the three
main tributaries (RES6-8). Stations RES1-3 which have the highest average bottom
concentrations are located in areas where the carbon density is 50% higher than the
agricultural ecosystems that were flooded in the area of the stations RES6-8 (Descloux et al.,

668 2011).

In the absence of significant vegetation regrowth in the drawdown area during the study
period, the main source of carbon fuelling emissions from the drawdown area are not clearly
identified. Immediately after flooding, the most labile part of the soil OM and the
decomposing vegetation must have been the main sources of C fuelling the emissions. On the
long haul, the atmospheric carbon sink associated with the pristine vegetation dynamics has
been lost but as a consequence, the loss of this vegetation which might reduce labile OM
inputs. In addition, the water level variations erode the soil and OM is transferred to the



reservoir and ultimately in the sediments or downstream (Félix-Faure et al., 2017). Those
carbon losses should have resulted or should result in the future in a decrease of $CO_2$
emissions from the drawdown. The stability of emissions throughout our four-years surveys
in the drawdown area suggests that new carbon source might have contributed to emissions.
Development of micro-phytobenthos or microbial biofilms as often observed in estuaries on
mudflats (de Brouwer and Stal, 2001) or along stream in logged riparian area (Sabater et al.,
2000) could supply labile OM and the system and favour priming effect (Guenet et al., 2010).
Through this effect, the inputs of labile OM stimulate the degradation/mineralization of
recalcitrant/stabilized OM. This effect might be enhanced by the oxic/anoxic oscillation that
would favour the mineralisation of different pool of OM than those that would have been
degraded otherwise in stable conditions (Abril et al., 1999;Bastviken et al., 2004). Overall, we
hypothesized that the oxic/anoxic variations and priming effect through the development of
algaes and bacteria might have contributed to the stability of $CO_2$ emission from these soils
under the influence of the reservoir. So far we found no clear evidence of a significant carbon
loss in the soils of the drawdown area by comparing surface SOM from pristine upland soils
and from the shoreline (Table 1). A comprehensive study of carbon density down to the
bedrock would be necessary since we found very clear evidence of inundation patterns down
to 1 m (P. Oliva, unpublished). In addition to the full carbon stock, detailed OM
characterisation might be needed for the identification of changes in the pool of soil OM.
The overall confirmation of the importance of the flooded pool of OM in the carbon cycling
in a tropical reservoir highlights the differences in functioning with boreal reservoirs where
the degradation of the flooded organic matter within a few year does not contribute
significantly to emissions (Brothers et al., 2012). In addition to a strong temperature effect on
mineralisation of OM (Gudasz et al., 2010), the probable low lability and good capacity for
preservation of peat-dominated OM might explain the different origin of OM fuelling
emissions between those two distinct climatic areas.
**5    Conclusions**
We presented the first comprehensive estimation of $CO_2$ emissions from a tropical reservoir
starting less than a year after reservoir impoundment and lasting 4.5 years. This estimation
includes all pathways to the atmosphere: emissions from the reservoir surface, downstream
emissions and emissions from the drawdown area.



More than 50% of total emissions occur within 3-4 months during the warmest period of the
year at the transition between the dry and the wet season. Such a result suggests that
quantification of emissions based on two to four campaigns in a year might significantly
affect positively or negatively emissions factors and carbon budgets of ecosystems under
study.
The smooth decrease of total emissions with time over the years coupled with the fact that the
incoming flux of carbon from the watershed to the reservoir represent less than a third of the
total emissions, are a strong indication that the flooded pool of organic matter is the main
source of carbon fuelling emissions. The carbon density of flooded soil and biomass in
reservoirs appears to be a key controlling factor of emissions and should be included for
future estimation of greenhouse gas emissions from reservoirs.
We found that gross $CO_2$ emissions from the drawdown area represented up to 75% of the
total emissions from the NT2R and they occur within a few months during low water level
seasons. The soil organic matter from these areas undergoes anaerobic degradation and fuels
the reservoir water column in $CO_2$ during the wet season. In the dry season, the soil loss $CO_2$
directly to the atmosphere while undergoing both aerobic and anaerobic mineralisation
depending on the soil moisture content. We hypothesize that both (1) the potential
development of bacteria and micro-phytobenthos at the surface of these soils and (2) the
oxic/anoxic variations contribute to the mineralisation of stabilized SOM leading to a
sustained high soil respiration even after the pristine vegetation decayed. This overlooked
pathway in terms of gross emissions would require an in-depth evaluation for the soil OM and
vegetation dynamics and long-term monitoring of emissions to evaluate the real contribution
of this area in terms of net modification of gas exchange in the footprint of the reservoir.
**Acknowledgements**
The authors thank everyone who contributed to the NT2 monitoring programme, especially
the Nam Theun 2 Power Company (NTPC), Electricité de France (EDF) and CNRS-INSU
(Submersoil project, EC2CO-BIOHEFECT) for providing financial, technical and logistic
support. We are also grateful to the Aquatic Environment Laboratory of the Nam Theun 2
Power Company whose Shareholders are EDF, Lao Holding State Enterprise and Electricity
Generating Public Company Limited of Thailand. CD benefited from a PhD grant by EDF.



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





Table 1 : Soil type and characteristics at the sampling station of the drawdown area of the Nam Theun 2 Reservoir (Lao PDR). KKK formation.

| Catena | solum | %N | %C | C:N | pH | Soil name WRB FAO | Soil texture | lithology |
|---|---|---|---|---|---|---|---|---|
| MNR | MNR upland | 0.11 | 1.47 | 13.69±1.73 | 4.33 | planosol | sandy | Micaceous quartzose |
|  | MNR interm. down | 0.10 | 1.32 | 13.55±1.84 |  | endogleyic planosol |  |  |
|  | MNR shoreline | 0.13 | 1.89 | 14.78±1.60 |  | gleysol |  |  |
| RES3S | RES3 upland | 0.18 | 2.38 | 13.21±0.63 | 4.18 | plinthosol | clay | Red mudstone |
|  | RES3 interm. |  |  |  |  | etagnic plinthosol |  |  |
|  | RES3 shoreline | 0.17 | 1.95 | 11.21±0.56 | 4.88 | plinthic stagnosol |  |  |
| RES2S | RES2 upland | 0.16 | 2.24 | 13.62±0.60 |  | plinthic ferralsol | sandy clay | Micaceous sandstone |
|  | RES2 interm. | 0.20 | 2.30 | 11.25±0.50 |  | « stagnic » ferralsol |  |  |
|  | RES2 shoreline | 0.13 | 1.41 | 10.55±0.54 |  | stagnosol |  |  |
| RES8S | RES8 upland | 0.08 | 1.76 | 23.47±3.95 |  | acrisol | sandy clay | Quaternary deposits |
|  | RES8 interm. up | 0.06 | 0.68 | 11.93±1.46 |  | stagnic acrisol |  |  |
|  | RES8 interm. down | 0.09 | 1.31 | 13.99±1.99 |  | stagnic acrisol |  |  |
|  | RES8 shoreline | 0.12 | 2.02 | 17.07±1.93 |  | endogleyic stagnosol |  |  |
| RES8'S | RES8' upland | 0.05 | 0.77 | 16.15±2.30 |  | acrisol | sandy clay | Quaternary deposits |
|  | RES8' shoreline | 0.08 | 1.51 | 18.22±2.79 |  | endogleyic stagnosol |  |  |
| RES4S | RES4 upland | 0.16 | 1.98 | 12.76±1.17 | 4.14 | acrisol | sandy clay | Micaceous sandstone |
|  | RES4 interm. up | 0.13 | 1.92 | 14.66±1.58 |  | stagnic acrisol |  |  |
|  | RES4 interm. down | 0.12 | 1.67 | 14.33±1.71 |  | stagnic acrisol |  |  |
|  | RES4 shoreline | 0.10 | 1.36 | 14.35±1.97 | 4.44 | gleysol |  |  |





Table 2 Temperature (°C), relative humidity (%) and $CO_2$ fluxes (mmol m$^{-2}$ d$^{-1}$) from the soils of the drawdown area of the Nam Theun 2 Reservoir (Lao PDR).

| Site | 2010 Hum | 2010 Temp | 2010 CO₂ flux | 2011 Hum | 2011 Temp | 2011 CO₂ flux | 2013 Hum | 2013 Temp | 2013 CO₂ flux |
|---|---|---|---|---|---|---|---|---|---|
| MNR upland | 17.5 | 25.7 | 265±37 | 18.3 | 24.4 | 328±43 | | | |
| MNR interm. up | | | | 26.9 | 27.5 | 669±56 | | | |
| MNR interm. down | 19.6 | 32.3 | 201±19 | 23.7 | 29 | 251±99 | | | |
| MNR shoreline | 37 | 31.9 | 40 | 46.4 | 27.3 | 67±7 | | | |
| RES3S upland | 22.3 | 26.8 | 231 | 23.6 | 25.6 | 366±14 | | | |
| RES3S interm. | 49.5 | 27.4 | 184±50 | 30.2 | 26.1 | 186±57 | | | |
| RES3 Sshoreline | 42.3 | 28.3 | 503±97 | 25.6 | 19.8 | 391±23 | | | |
| RES2S upland | 19.9 | 26.4 | 183±1 | 24.5 | 25.2 | 531±41 | | | |
| RES2S interm. | 34.6 | 29.2 | 138±21 | 30.2 | 26.1 | 339±52 | | | |
| RES2S shoreline | 49.4 | 28.5 | 332±5 | 48.7 | 27.1 | 166±23 | | | |
| RES8S upland | 27.7 | 28.2 | 86±0 | 26.9 | 27.0 | 468 | | | |
| RES8S interm. up | 32.3 | 28.3 | 75±15 | 33.2 | 26.9 | 300±19 | | | |
| RES8S interm. down | 32.9 | 29.1 | 110±10 | 32.3 | 27.8 | 239±44 | | | |
| RES8S shoreline | 45.3 | 29.7 | 286±59 | 44.5 | 28.5 | 660±121 | | | |
| RES8S' upland | 32.6 | 32.5 | 342±70 | | | | | | |
| RES8S' interm. | 35.9 | 31.9 | 143±24 | | | | | | |
| RES8S'shoreline | 42.7 | 31.9 | 34±7 | | | | | | |
| RES4S upland | 26.7 | 28.6 | 326±20 | 21.7 | 29.5 | 526±35 | 18.1 | 31.1 | 232±50 |
| RES4S interm. up | | | | | | | 24.3 | 28.7 | 196±29 |
| RES4S interm. down | 26.0 | 34.2 | 168±28 | 21.8 | 32.7 | 619±39 | 35.1 | 29.8 | 443±67 |
| RES4S shoreline | 44.6 | 31.1 | 34±7 | 18.3 | 32.1 | 115 | 51.2 | 29.6 | 393±57 |





Table 3: $CO_2$ emissions (in $GgCO_2.year^{-1}$) from the Nam Theun 2 Reservoir (Lao PDR) for the first five years after impoundment (2009, 2010, 2011, 2012 and 2013). Percentages between brackets represent the proportion of each component to the total annual emission.

| Year | Ebullition | Diffusion (Reservoir) | Diffusion (Drawdown) | Degassing | Diffusion (Downstream) | Total |
|---|---|---|---|---|---|---|
| 2009 | 1.2±0.5 (<1%) | 730.0±46.2 (92%) | 6.3±0.5 (1%) | 52.7±14.9 (7%) | 4.0±0.3 (<1%) | 794.1±48.5 |
| 2010 | 1.04±0.5 (<1%) | 538.57±28.6 (51%) | 413.7±15.9 (39%) | 85.37±17.4 (8%) | 14.34±0.4 (1%) | 1053.0±37.0 |
| 2011 | 1.06±0.5 (<1%) | 345.88±24.3 (42%) | 386.4±16.0 (47%) | 84.03±10.7 (10%) | 11.60±0.5 (1%) | 828.9±31.0 |
| 2012 | 0.95±0.4 (<1%) | 173.30±11.5 (23%) | 572.3±19.9 (75%) | 17.03±3.8 (2%) | 2.23±0.2 (<1%) | 765.8±23.3 |
| 2013 | 1.04±0.5 (<1%) | 118.70±27.3 (21%) | 419±15.0 (76%) | 13.61±4.0 (2%) | 1.43±0.2 (<1%) | 553.8±31.4 |





Figure 1 Map of the Nam Theun 2 monitoring network



Figure 2 : Median and interquartile range (boxes), average (+), and full range of values (whiskers) of particulate organic carbon (POC), dissolved organic carbon (DOC), total inorganic carbon (TIC) and CO2 concentrations in four pristine river of the Nam Theun watershed during three distinct seasons : cold dry (CD), warm dry (WD) and warm wet (WW) seasons. The dataset includes data from 2009 to 2013.

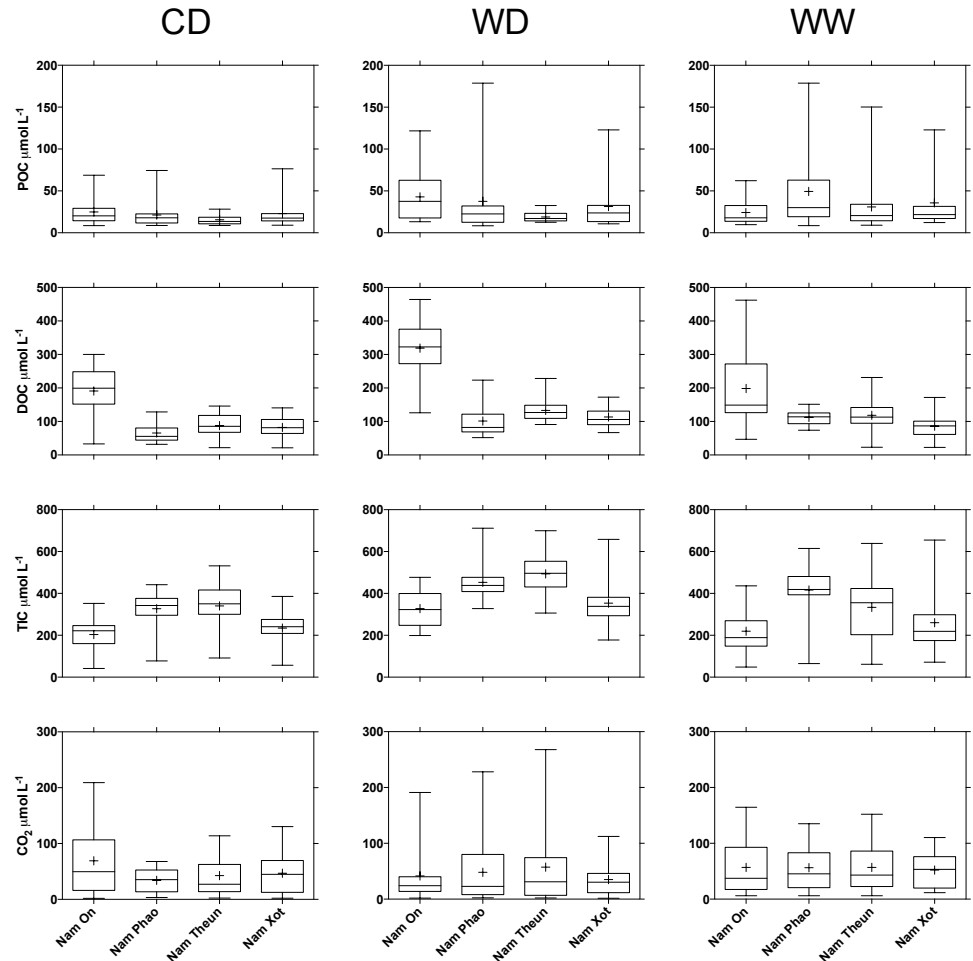



Figure 3: Total carbon inputs in form of particulate organic carbon (POC), dissolved organic carbon (DOC) and total inorganic carbon (TIC) from the Nam Theun watershed to the Nam Theun 2 Reservoir for four distinct years after reservoir impoundment.

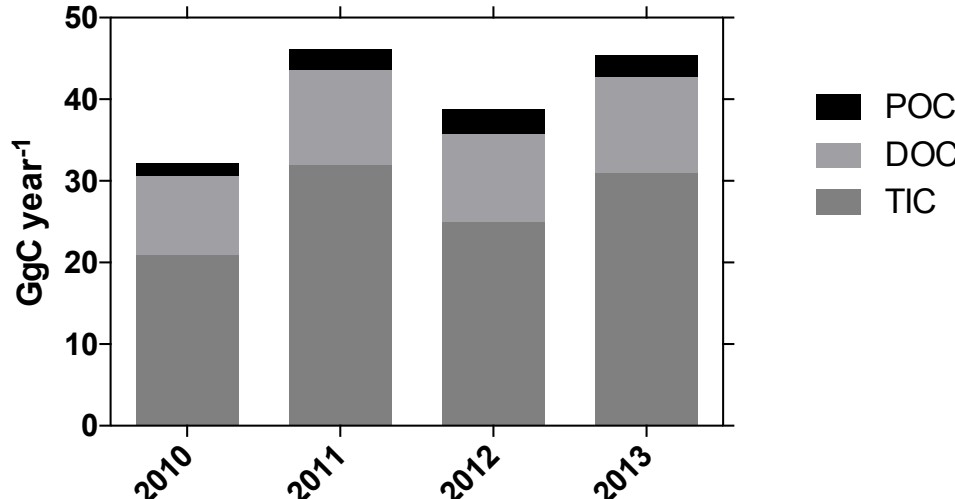



Figure 4: Temperature (grey solid circle) and oxygen (black solid circle), DOC (open square), POC (solid square) and $CO_2$ (triangle) concentrations in the Nam Theun 2 Reservoir water column during the cool dry, warm dry and warm wet seasons in 2011 at three stations (RES3, RES7 and RES9).

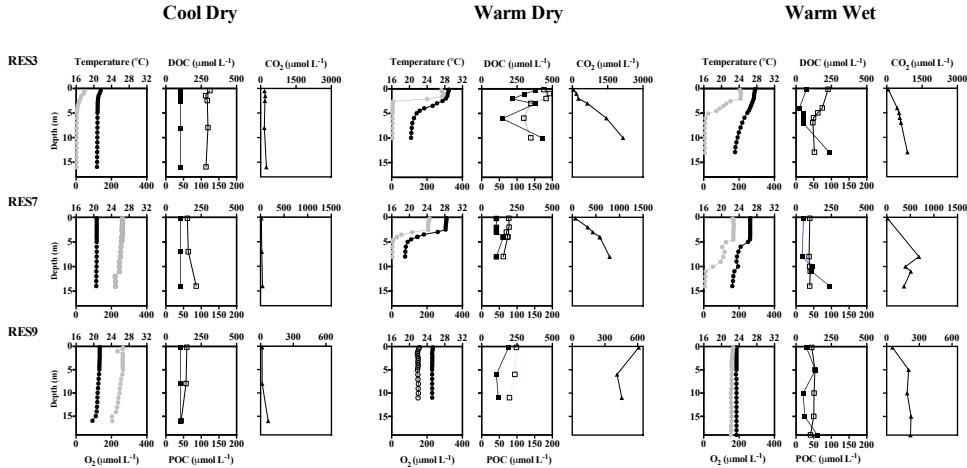



Figure 5: (a) Monthly average $CO_2$ concentrations at the stations RES1-8 (a) and at the station RES9 (b), average diffusive fluxes at the stations RES1-8 (c) and at the station RES9 (d) and total monthly (e) and yearly (f) $CO_2$ emissions by diffusive fluxes from the Nam Theun 2 Reservoir (Lao PDR)

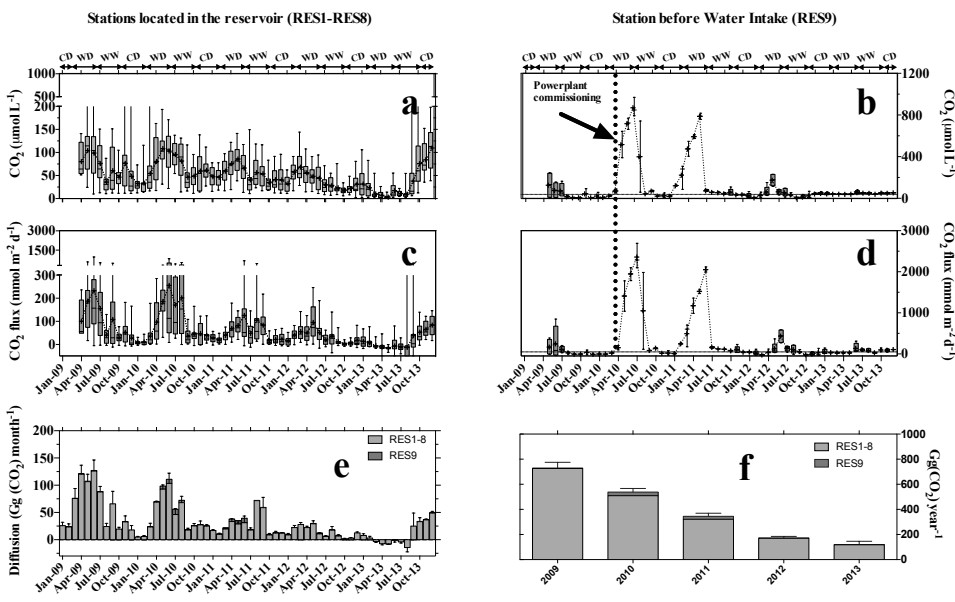



Figure 6: Diffusive fluxes and degassing below the powerhouse and the Nakai Dam on a monthly (a) and yearly basis (b) at the Nam Theun 2 Reservoir (Lao PDR). Note that degassing below ND includes spillway release (main contributor to 2009 and 2011 emissions below ND). Degassing below the powerhouse includes degassing immediately downstream of the turbines, downstream of the regulation dam and downstream of the aeration.

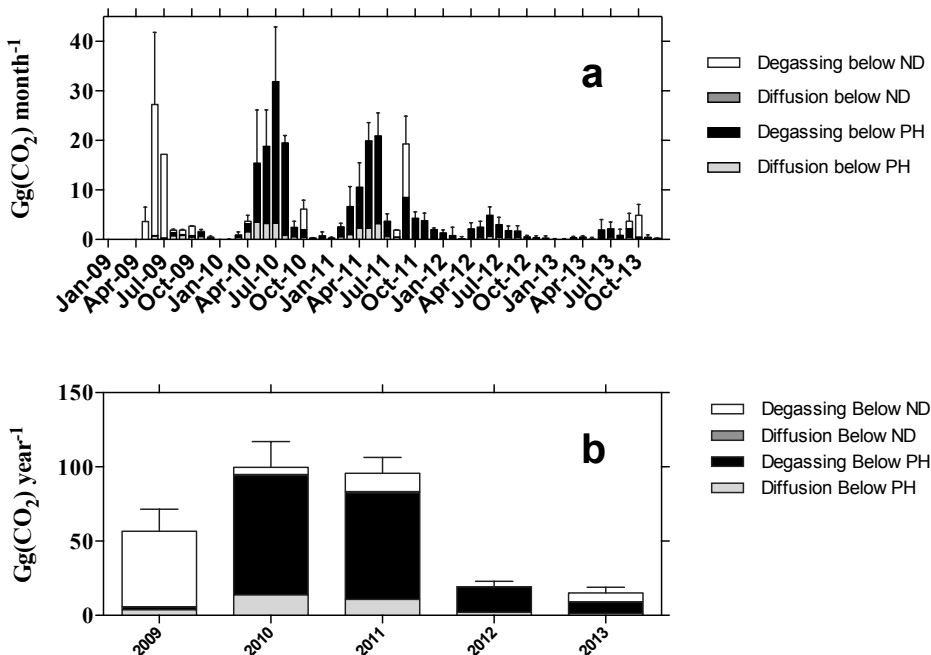



Figure 7: Monthly emissions from the drawdown area and variation of the area of the drawdown area of the Nam Theun 2 Reservoir (Lao PDR)

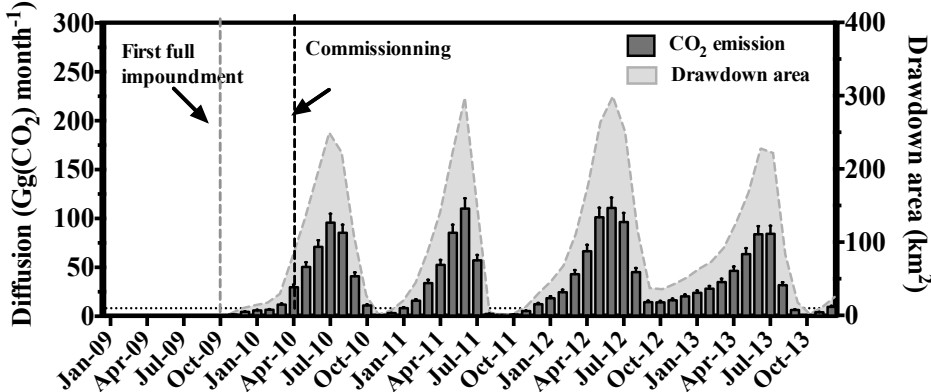




Figure 8: Monthly (a) and yearly (b) average of the total emissions from the Nam Theun 2 Reservoir by diffusion at the reservoir surface, diffusion from the drawdown area, ebullition, degassing and diffusion from the Nam Theun River and artificial channel at the Nam Theun 2 Reservoir (Lao PDR). On panel a, water level variations in the reservoir are given.

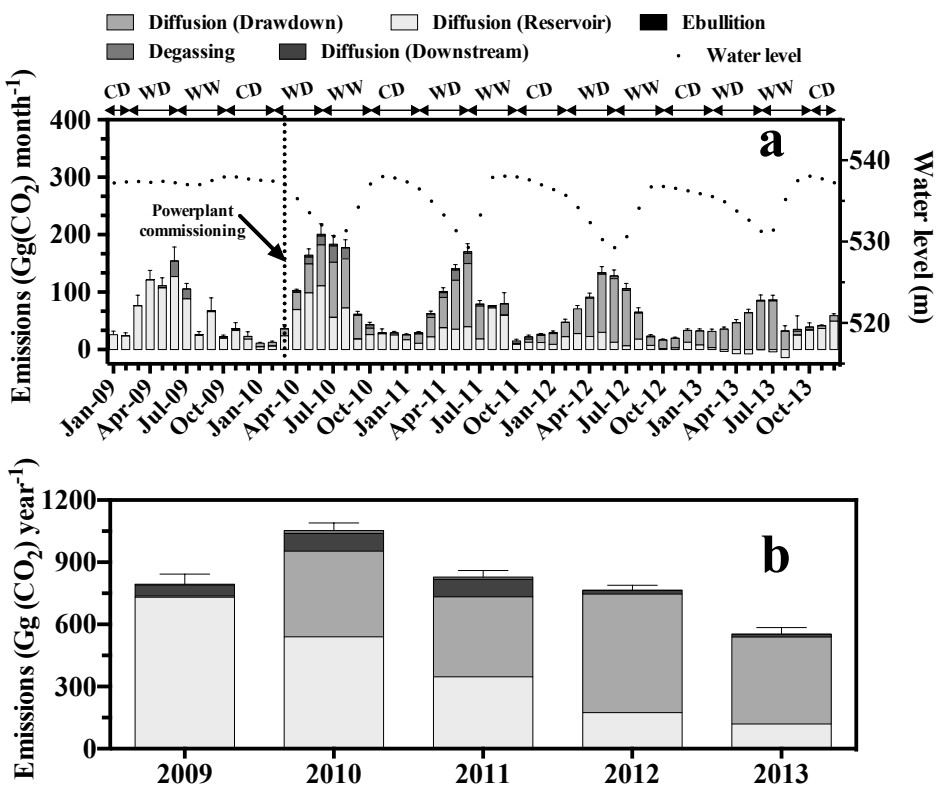