# Peer review of "Nam Theun 2 Reservoir: drawdown area as a neglected"

_Biogeosciences, 2017_

## Referee Comment (RC1) · Anonymous Referee #1 · 2 Nov 2017

The manuscript presented results of a study carried out on carbon dioxide emissions from the Nam Theum 2 Reservoir in the Mekong River watershed in Laos. The major focus has been on the influence of Dam and commissing of a power plant. Their study clearly shows the impact of human interference on the natural flow systems and processes on carbon dioxide system and its emissions. The authors deserve compliments for their meticulous planning of their experiments and strategic location of sampling sites. Results are presented and discussed adequately and I do not have any major comments. The following three minor points may help the authors in contributing

to the clarity.

1. Please define 'drawdown area' in introduction. 2. Caption for Figure 5 needs a revision and 3. The influence of freshwater discharge on distributions of carbon parameters in the studied hydrological regime has not been explicitly presented. A strategy showing the river discharge variations in relation to changes in carbon dioxide properties in reservoir and drawdown area might help explain "We confirm the importance of the flooded stock of organic matter as a source of C fuelling emissions and we show that the drawdown area contributes, depending on the year, from 50% to 75% of total annual gross emissions in this flat and shallow reservoir (lines 47-50)."

---

## Referee Comment (RC2) · NSS Sarma (Referee) · 7 Nov 2017

General: The contribution of global freshwater reservoirs to the atmospheric CO2 is an important problem. Although the storage bodies, the reservoirs proper have been examined in reasonable detail, emissions in the downstream regions adjacent to the dams in the flow paths have not been addressed sufficiently. In this background, the present paper is welcome. The authors previously published in the same journal (Biogeosciences) on CH4 emissions, as 2 papers, the first one dealing with downstream stations (Deshmukh et al., 2016) and the second dealing with the reservoir proper

(Guelin et al., 2016). This MS is on CO2 emissions for the combined area. The experimental work is solid strong and data of high quality. However, after reading their 2 papers also (along with the present), the sampling protocol, flux calculations and discussion of results are much the same. The readers would be justified to expect from this paper not just about concentrations and fluxes of CO2, but a critical appraisal, in particular differences between CH4 and CO2 and a geochemical reasoning in terms of the processes / geochemistry. To a reader with taste for science, the Results and Discussion appeared routine, unnecessarily long and repetitive. The authors, during discussion (L. 553/557) did briefly mention about the differences in concentration trends of CH4 and CO2 but did not go further as to explain the processes except to mention that higher solubilization of CO2 leads to higher concentration. CO2 indeed provides a greater opportunity to discuss its more complex environmental response than CH4. CO2 is a reactive gas, unlike CH4 which undergoes only physical dissolution. CO2's reaction with water produces HCO3- , CO32-, H2CO3 in addition to physically dissolved CO2(aq) species all of which inter-convert as part of the carbonate equilibria. Due to the pH dependence of their inter-conversion, CO2(aq) and HCO3- are ∼50% each at pH 6 while at pH 10, HCO3- and CO32- are ∼50% each. At lower pH, degassing is favoured which happens in 2 cases, (i) seasonally in winter when the reservoir experiences overturning and (ii) spatially at the reservoir station 9 where mixing with the low pH deepwater takes place. The pH which varied significantly – in different ranges at different stations / regions may be reflecting these processes. In the reservoir and at various other water stations pH varied significantly. For example, at reservoir surface, the range was 5.21 - 8.76 (L. 271) when the corresponding share of CO2(aq) in the CO2 system is >80% and ∼10% respectively, and the former situation is a hugely favourable CO2 emission condition. Post degassing, pH should be expected to increase at surface (up to the limit of neutral pH). But the higher limit of pH which was on the alkaline side (pH>7) shows that there are cations (from dissolved minerals) e.g., Na+, K+ etc whether derived naturally or anthropogenically. In addition to CO2 (aq), authors measured TIC, but they did not explore CO2 emission in relation to the TIC-

CO2(aq) equilibrium leading to the basic question as to why they presented the latter data. Discussion of Figs. 2 and 3 is absent except for a brief mention of the relative quantities / fluxes of DOC, POC and TIC. For CO2 and TIC determination, authors gave citations of their earlier works. It would be useful if the methods are explained in brief. Production and accumulation of CO2: Authors have not explained how. Using water residence time and vertical stratification index authors explained in their papers (e.g. Guerin et al., 2016). They also could relate CO2 production (by the metabolism of organic matter of sediments and water column by bacteria) and accumulation to age. The deep water is more aged than the surface water, and in it CO2 accumulated over longer periods also resulting in lower pH. The detailed hydrology and minor variations in concentrations should all fall in pattern if this were done. Thus, authors have to better consider a process-oriented description of their results rather than a just presentation of concentrations and fluxes. Further comments: General: 1. The CO2 concentrations (Text e.g., L. 394, 396, 428 etc.) and emissions (Table 3) are given in grams. The standard method is to give them in terms of CO2-C. The values would then be down by a factor of 44/12 i.e., 3.67. 2. Please give a space after semicolon (;) for all multiple citations. 3. L. 73: drawdown emissions: To my understanding, draw-down is opposite of emission. The former is from atmosphere to surface water when surface water is under-saturated (this is promoted by primary production) and the latter is from the surface water to the atmosphere in case of surface super-saturation (this is promoted by winter convection, which you are calling as reservoir overturning). There is no mention of drawdown emissions in for example Chen et al., 2009 cited by you. Do you mean emission in the drawdown area i.e., the reservoir or river area where the water level is lowered due to the construction of the reservoir? If so, the drawdown emissions should be replaced with emission in drawdown area throughout the MS. 4. Often, the results are specific to only the study area, and not applicable as a general phenomenon which makes the reading less involving for the reader. Hence, the authors better discuss critically their results focusing on (i) similarities and (ii) differences with other similar reservoirs. In the Discussion section, attention may be paid to spatial differences and

seasonal differences in sub-sections. 5. Fig. 8a constitutes the core result, and instead of waiting till the end of discussion, this figure may be brought to Results section, and later discussed critically (in the light of relevant comments below). 6. A significant part of discussion draws on CH4 distribution, but a direct comparison of the two results is not made. The drawdown area is an important source of CH4 7. Let me also give my opinion on the Title: As commented above, emissions from the drawdown area are significant only during the warm season when the drawdown area is exposed with fall in water level. Moreover, there is a gradual fall in these emissions too. Perhaps, if the dam were visited in 2017, the emissions may be expected to be further low, which is also mentioned by authors (L. 61-61 and 632-634). Hence, it may be misleading to say that drawdown areas are a neglected pathway to the atmosphere. 8. Interestingly, CH4 emission also took place during the dry season and the authors (Deshmukh et al., 2016) explained it to be due to intermittent exposure (and inundation) when anoxic (and oxic) conditions prevailed. Perhaps this point in itself would suggest the need for a direct comparison of the CO2 and CH4 results. Specific: L. 35: Pl. include in Laos PDR before in the Mekong River water shed. L. 39-40: Where are the river stations (Nam Theun watershed) in Fig. 1? Should there be a comma after Nam Theun water-shed in Line 40? L. 40: Nine: Change to 9 for consistency. L. 44: in 2012-2013: Pl. change to during 2013-2013, as monitoring was done in both years. L. 77: Pl. add in China before the citation. L. 104: decreased down to 107 km2: from what area? Is it about 500 km2? L. 107: m3s-2: This is not a correct unit for discharge. Later you mentioned m3s-1 which is right. L. 123-125: besides the hydrology details which were already described in Guerin et l. (2016), it would be good if you can give depths of the stations also. L. 159: What is specific water discharge? L. 197: soils types: Pl. correct to soil types L. 199: details: Pl. use singular (detail) as above. And pl. make similar corrections elsewhere also. L. 199: Table 1 – what is interm. for? L. 213: One of the subsample: Pl. correct it as one of the subsamples (Pl. compare with the above two corrections). L. 221: What is specific water discharge? What is Hum? L. 236: In Fig. 2, it would be better if the data are provided for the area classification followed in Fig.

[Figure]

8. L. 255 (also L. 638): This data has not been critically discussed. L. 259: This figure is illegible. The trends are not clearly seen due to the problem of scaling of the X-axis. L. 300: Are 70% and 56% (for 2011 and 2010 & 2012) annual average O2 saturation values or seasonal values? Pl. clarify. Pl. modify text for better clarity. L. 301: the is a repetition. L. 325: From March to August: You have referred so far to seasons. Better be consistent and refer as WD and WW seasons. L. 332: space between five and fold. L. 338: Pl. change a to an. L. 337-340: Why was this? Pl. explain in Discussion. L. 342: Pl. use on instead of to. L. 344: Figure 5e: These are also the trends shown in 5c. Suggest removing. Suggest removing 5f also as this data is given in Table 3 (column 3). L. 390: -32-33762: Pl. clarify the hyphen. The first hyphen seems to be a negative sign and the latter for range. L. 391: (not shown): The data can be included in Figure 5 as replacements for 5e and 5f to be deleted (see an earlier comment). L. 401: Fig. 6b: This data are included in Table 3 (column 5). Pl. remove. L. 419: Where is Figure 6d? L. 424-425: no bubbles was ever observed for depth higher than 16 m: Pl. delete text as this was given in Methods section. L. 434: stagnic property: Pl. explain briefly what a stagnic property is. L. 439-440: surface moisture ranging from 17.5 to 51.2% and temperature ranging from 18.1 to 34.2°C (Table 2): For consistency, pl. change text as: surface moisture (17.5 - 51.2%) and temperature (18.1 -34.2°C) (Table 2). L. 443: This p value of 0.452 is not significant! Is it a typo? L. 443-445: This sentence is not self-explanatory. L. 449: could reach: For consistency, pl. change to reached. L. 450, 451: Pl. change changes to changed. L. 454-455: Fig. 7 indicates that 2012 emissions were higher as July and August were also CO2-emitting. Pl. explain why under Discussion. L. 473-475: This sentence is a repetition of the earlier sentence in content. L. 475: This assumption is reinforce: Pl. correct to This assumption is reinforced L. 475: hot moments: When were those hot moments and why? L. 477: the higher concentrations were observed: Pl. remove the definite article. Also, explain why. L. 481: Pl. change the first of to a L. 482: change was observed nutrient concentrations: Correct to change was observed in nutrient concentrations. L. 487-489: No, the quantity of autochthonous OM is not greater than phytoplankton

primary production. Hence, there should be some other mechanism (source). L. 498: older reservoir: Pl. change to older reservoirs. Fig. 1: This figure is cluttered. The station codes are too long (and also not explained) and contribute to this clutter. What is the direction of river flow? What are NKT, TRC, DCH and XBF? The artificial channel is not marked properly in figure, and it is difficult to understand when mentioned e.g. in L. 530. Some terms included in legend e.g., Stream and downstream channel occur nowhere in text. Res 1 and downstream of reservoir – are they same? It should help the reader if you explained the provenance of different sampling stations in the Methods section, or as commented under Table 3. L. 549: For consistency, pl. change between the WD and the WW season (April – July). L. 551: emissions factors: Pl. change to emission factors. L. 552-557: This difference between CH4 (earlier work) and CO2 could be explored further. L. 555: Table 3: Pl. give data separately for Res 9. This can be done by inserting a row after the header row for giving the stations included. L. 559: compare to most of the reservoirs: Pl. correct as compared to most of the reservoirs. L. 564-567: This sentence is a repetition from earlier discussed. L. 568: For consistency of tense, pl. change increase to increased. L. 568-569: This sentence is also a repetition (Pl. see the opening sentence of this Section!). L. 579: down to 7oC in air in March 2011: Was this given under Results? L. 620: were taking into: Pl. correct as were taken into L. 624: this study highlights: But this study is about CO2 only. L. 688: Pl. correct algaes as algae L. 696: in a tropical reservoir: Pl. specify. Correct it as in the tropical NAM 2 reservoir. L. 705: Pl. change all with different L. 708-711: This sentence is redundant. L. 712: with time over the years: Pl. remove with time. L. 713: represent: Pl. correct it to represents. L. 715-717: But this is important only in the initial years after impoundment as evident in Fig. 8a. By the year 2013, the emissions have decreased significantly. Also, during the WW and particularly CD season, a seasonal shift of emissions happened and the reservoir emissions far surpassed the emissions from the drawdown area, thereby restoring the condition existed pre-power plant commissioning. Thus, the drawdown area and Reservoir have their own seasons when emissions peak – WD and the initial part of WW seasons in the

former and the later part of WW season and CD season in the case of the latter. Pl. explain clearly under Discussion the result and why it is so. L. 718-720: Although the % emissions from the drawdown area is 75% of total, in absolute terms, the emission (quantity) is same or perhaps less in 2013, as prior to commissioning. L. 729: footprint of the reservoir: What is footprint? Not discussed earlier under Discussion.

I am of the opinion that the paper requires major revision.

Please also note the supplement to this comment:
https://www.biogeosciences-discuss.net/bg-2017-380/bg-2017-380-RC2-supplement.pdf

―――――――――――――――――――

---

## Author Comment (AC1) · 13 Jan 2018

The authors thank the reviewer for her/his positive comments on our work and for her/his constructive remarks

The manuscript presented results of a study carried out on carbon dioxide emissions from the Nam Theum 2 Reservoir in the Mekong River watershed in Laos. The major focus has been on the influence of Dam and commissioning of a power plant. Their study clearly shows the impact of human interference on the natural flow systems and processes on carbon dioxide system and its emissions. The authors deserve compli- ments for their meticulous planning of their experiments and strategic location of sam- pling sites. Results are presented and discussed adequately and I do not have any major comments. The following three minor points may help the authors in contributing to the clarity.

1. Please define 'drawdown area' in introduction.
The following lines were added in the introduction: "In some reservoirs with large water level variations, large surface areas of soils known as drawdown zones are periodically exposed to the atmosphere (for example, Three- Gorges and Nam Theun 2 reservoirs)."

2. Caption for Figure 5 needs a re- vision
The extra (a) label was removed from the caption

and 3. The influence of freshwater discharge on distributions of carbon param- eters in the studied hydrological regime has not been explicitly presented. A strategy showing the river discharge variations in relation to changes in carbon dioxide proper- ties in reservoir and drawdown area might help explain "We confirm the importance of the flooded stock of organic matter as a source of C fuelling emissions and we show that the drawdown area contributes, depending on the year, from 50% to 75% of total annual gross emissions in this flat and shallow reservoir (lines 47-50)."

The large variation of the contribution of the drawdown zone do not result from hydrological variations since emissions from the drawdown zone are constant throughout the years. The following sentence was added in order to clarify: "Since the CO2 emissions from the drawdown zone are almost constant throughout the years, the large interannual variations result from the significant decrease of diffusive fluxes and downstream emissions between 2010 and 2013."

---

## Author Comment (AC2) · 13 Jan 2018

The authors thank the reviewer for his thorough review of the manuscript

General: The contribution of global freshwater reservoirs to the atmospheric CO2 is an important problem. Although the storage bodies, the reservoirs proper have been examined in reasonable detail, emissions in the downstream regions adjacent to the dams in the flow paths have not been addressed sufficiently. In this background, the present paper is welcome. The authors previously published in the same journal (Biogeosciences) on CH4 emissions, as 2 papers, the first one dealing with downstream stations (Deshmukh et al., 2016) and the second dealing with the reservoir proper (Guerin et al., 2016). This MS is on CO2 emissions for the combined area. The experimental work is solid strong and data of high quality. However, after reading their 2 papers also (along with the present), the sampling protocol, flux calculations and discussion of results are much the same. The readers would be justified to expect from this paper not just about concentrations and fluxes of CO2, but a critical appraisal, in particular differences between CH4 and CO2 and a geochemical reasoning in terms of the processes / geochemistry. To a reader with taste for science, the Results and Discussion appeared routine, unnecessarily long and repetitive. The authors, during discussion (L. 553/557) did briefly mention about the differences in concentration trends of CH4 and CO2 but did not go further as to explain the processes except to mention that higher solubilization of CO2 leads to higher concentration.

We have already published 5 papers on CH4 emissions from Nam Theun 2 Reservoir (see below) and we do not feel that new discussion on CH4 is necessary. The present manuscript focuses on CO2 emissions from the major known pathways and we demonstrate for the first time the existence of an overlooked pathway i.e. the drawdown area, which constitutes, in our opinion, a significant result worth a stand-alone paper.

Descloux, S., V. Chanudet, B. Taquet, W. Rode, P. Guédant, D. Serça, C. Deshmukh and F. Guérin (2016). "Efficiency of the Nam Theun 2 hydraulic structures on water aeration and methane degassing." Hydroécol. Appl. 19: 63-86.

Deshmukh, C., F. Guérin, D. Labat, S. Pighini, A. Vongkhamsao, P. Guédant, W. Rode, A. Godon, V. Chanudet, S. Descloux and D. Serça (2016). "Low methane (CH4) emissions downstream of a monomictic subtropical hydroelectric reservoir (Nam Theun 2, Lao PDR)." Biogeosciences 13(6): 1919-1932.

Deshmukh, C., D. Serca, C. Delon, R. Tardif, M. Demarty, C. Jarnot, Y. Meyerfeld, V. Chanudet, P. Guedant, W. Rode, S. Descloux and F. Guérin (2014). "Physical controls on CH4 emissions from a newly flooded subtropical freshwater hydroelectric reservoir: Nam Theun 2." Biogeosciences 11(15): 4251-4269.

Guérin, F., C. Deshmukh, D. Labat, S. Pighini, A. Vongkhamsao, P. Guédant, W. Rode, A. Godon, V. Chanudet, S. Descloux and D. Serça (2016). "Effect of sporadic destratification, seasonal overturn, and artificial mixing on CH4 emissions from a subtropical hydroelectric reservoir." Biogeosciences 13(12): 3647-3663.

Serça, D., C. Deshmukh, S. Pighini, P. Oudone, A. Vongkhamsao, P. Guédant, W. Rode, A. Godon, V. Chanudet, S. Descloux and F. Guérin (2016). "Nam Theun 2 Reservoir four years after commissioning: significance of drawdown methane emissions and other pathways." Hydroécol. Appl. 19: 119-146.

CO2 indeed provides a greater opportunity to discuss its more complex environmental response than CH4. CO2 is a reactive gas, unlike CH4 which undergoes only physical dissolution. CO2's re- action with water produces HCO3- , CO32-, H2CO3 in addition to physically dissolved CO2(aq) species all of which inter-convert as part of the carbonate equilibria. Due to the pH dependence of their inter-conversion, CO2(aq) and HCO3- are ~ 50% each at pH 6 while at pH 10, HCO3- and CO32- are ~50% each. At lower pH, degassing is favoured which happens in 2 cases, (i) seasonally in winter when the reservoir expe- riences overturning and (ii) spatially at the reservoir station 9 where mixing with the low pH deepwater takes place. The pH which varied significantly – in different ranges at different

stations / regions may be reflecting these processes. In the reservoir and at various other water stations pH varied significantly. For example, at reservoir sur- face, the range was 5.21 - 8.76 (L. 271) when the corresponding share of CO2(aq) in the CO2 system is >80% and ~ 10% respectively, and the former situation is a hugely favourable CO2 emission condition. Post degassing, pH should be expected to in- crease at surface (up to the limit of neutral pH). But the higher limit of pH which was on the alkaline side (pH>7) shows that there are cations (from dissolved minerals) e.g., Na+, K+ etc whether derived naturally or anthropogenically. In addition to CO2 (aq), authors measured TIC, but they did not explore CO2 emission in relation to the TIC-CO2(aq) equilibrium leading to the basic question as to why they presented the latter data.

As in the majority of lakes and reservoirs, CO2 in produced throughout the water column by aerobic or anaerobic respiration and is partly consumed in the euphotic zone by primary production. Mineralization of organic matter through bacterial activity lead to CO2 production which acidifies the environment and to direct production of protons while consumption of CO2 by primary production increase the pH of surface water during productive periods. This is classical vertical profiles of pH in such environments.
TIC results are mostly used for the carbon mass balance and the comparison of total input from the watershed with the total export downstream and emissions to the atmosphere. This comparison is the basis of the section 4.3 in the discussion. As TIC dominates the carbon inputs to the reservoir, it is a key element of the article.

Discussion of Figs. 2 and 3 is absent except for a brief mention of the relative quantities / fluxes of DOC, POC and TIC.
The figure 2 depict raw data (TIC, DOC, POC, CO2) in all rivers from the Nam Theun watershed that were used for the calculation of carbon inputs from the Nam Theun watershed presented in figure 3.
The section 4.3 on the source of carbon fuelling emissions in the NT2 reservoir is based on the comparison of figure 3 showing the carbon inputs from the watershed and the figure 8 showing emissions from the NT2 reservoir. As mentioned above, carbon inputs from the watershed are key elements of the mass balance and discussion.

For CO2 and TIC determination, authors gave citations of their earlier works. It would be useful if the methods are explained in brief.
The headspace method used for CO2 measurements is well known and the name of the method is self-explicit (eg, Guérin et al., 2006). TIC, DOC, TOC measurements are routine measurements with a TOC analyser from shimadzu as done in numerous published studies. We believe the description made in the article for the sample preparation and analysis is detailed enough.

Production and accumulation of CO2: Authors have not explained how. Using water residence time and vertical stratification index authors explained in their papers (e.g. Guerin et al., 2016). They also could relate CO2 production (by the metabolism of organic matter of sediments and water column by bacteria) and accumulation to age. The deep water is more aged than the surface water, and in it CO2 accumulated over longer periods also resulting in lower pH. The detailed hydrology and minor variations in concentrations should all fall in

pattern if this were done. Thus, authors have to better consider a process-oriented description of their results rather than a just presentation of concentrations and fluxes.

The authors do not understand the point raised by the reviewer. The surface water is less concentrated in CO2 and has a higher pH because of primary production and loss of CO2 by emissions. The water in the reservoir comes from the watershed and difference in age between surface and bottom water are not expected in a reservoir. Some parts of reservoir experiences longer residence time than other parts, and this is places where CO2 concentration are higher (see L546-548) We observed slightly lower pH, but these differences were not significant..

Further comments:
General:
1. The CO2 concentrations (Text e.g., L. 394, 396, 428 etc.) and emissions (Table 3) are given in grams. The standard method is to give them in terms of CO2-C. The values would then be down by a factor of 44/12 i.e., 3.67.

There is no official recommendation for Biogeosciences. There are as many published studies reporting fluxes in gC-CO2 as fluxes in gCO2. We kept our data in gCO2.

2. Please give a space after semicolon (;) for all multiple citations.

The absence of space is due to the use of the Endnote software. This would be solved during the manuscript processing

3. L. 73: drawdown emissions: To my understanding, draw-down is opposite of emission. The former is from atmosphere to surface water when surface water is under-saturated (this is promoted by primary production) and the latter is from the surface water to the atmosphere in case of surface super-saturation (this is promoted by winter convection, which you are calling as reservoir overturning). There is no mention of drawdown emissions in for example Chen et al., 2009 cited by you. Do you mean emission in the drawdown area i.e., the reservoir or river area where the water level is lowered due to the construction of the reservoir? If so, the drawdown emissions should be replaced with emission in drawdown area throughout the MS.

It seems that the reviewer is not familiar about what is a drawdown area in a reservoir. This term is widely spread nowadays in the framework of GHG emissions from hydroelectric reservoirs studies. As an example, the paper by Chen et al 2009 is namely focused in that very precise point as illustrated in the title: *"Chen, H., Y. Wu, X. Yuan, Y. Gao, N. Wu and D. Zhu (2009). "Methane emissions from newly created marshes in the drawdown area of the Three Gorges Reservoir." J. Geophys. Res. 114: D18301."*
We do not understand what the reviewer is refering to when he writes that "the creation of a reservoir would lower the water level in the area".
Since the focus of our paper is on the drawdown area of a monomictic reservoir (a reservoir which overturn once a year), we hardly see how the reviewer's comment can be considered as relevant.

4. Often, the results are specific to only the study area, and not applicable as a general phenomenon which makes the reading less involving for the reader. Hence, the authors better discuss critically their results focusing on (i) similarities and (ii) differences with other

similar reservoirs. In the Discussion section, attention may be paid to spatial differences and seasonal differences in sub-sections.

Results are by definition specific to the area. We always bring comparison with other sites as it can be seen in almost all paragraphs of the discussion. Last paragraph of each section attempts to generalise the results to other sites or climatic zones whenever it is relevant.

5. Fig. 8a constitutes the core result, and instead of waiting till the end of discussion, this figure may be brought to Results section, and later discussed critically (in the light of relevant comments below).

The Fig 8 cannot be shown before all individual terms of emissions are described and discussed in details in terms of spatial and temporal variability in the result section. It makes sense that the figure 8, which is the synthesis of all results, is referred to in the discussion section where it is commented in details.

6. A significant part of discussion draws on CH4 distribution, but a direct comparison of the two results is not made. The drawdown area is an important source of CH4

CH4 is cited 9 times in the discussion and conclusion which correspond to 9 pages, it is therefore not a significant part of the discussion and as mentioned in our first comment, our CH4 dataset is published in a 5 papers and does not deserve more attention, specially in a paper focusing on specific pathway for CO2.
We clearly demonstrated in Serca et al., 2016 that the drawdown area of this reservoir is not a significant source of CH4 (3% of total emissions).

7. Let me also give my opinion on the Title: As commented above, emissions from the drawdown area are significant only during the warm season when the drawdown area is exposed with fall in water level. Moreover, there is a gradual fall in these emissions too. Perhaps, if the dam were visited in 2017, the emissions may be expected to be further low, which is also mentioned by authors (L. 61-61 and 632-634). Hence, it may be misleading to say that drawdown areas are a neglected pathway to the atmosphere.

Based on common definition of drawdown zone, this title clearly justifies the outcomes of this research work.. According to Abril et al. (2005) at Petit Saut, total emissions (disregarding drawdown emissions which were not measured) were 9 to 6 times higher than carbon inputs from the watershed during the first 4 years for similar carbon inputs which indicates a faster decrease of emissions in NT2R than at Petit Saut. This sharp decrease of emissions at NT2R might be due to the fact that the flooded pool of OM and therefore the amount of labile OM in NT2R was twice smaller than the amount of OM flooded in the Petit Saut (Guérin et al., 2008;Descloux et al., 2011).

8. Interestingly, CH4 emission also took place during the dry season and the authors (Deshmukh et al., 2016) explained it to be due to intermittent exposure (and inundation) when anoxic (and oxic) conditions prevailed. Perhaps this point in itself would suggest the need for a direct comparison of the CO2 and CH4 results.

Deshmukh et al (2016) deals with downstream emissions and not with drawdown area as the reviewer understood.

Specific:
L. 35: Pl. include in Laos PDR before in the Mekong River water shed.

Lao PDR was added

L. 39-40: Where are the river stations (Nam Theun watershed) in Fig. 1? Should there be a comma after Nam Theun water- shed in Line 40?
Pristine was added to the sentence for consistency with the Fig 1 caption and a coma was added

L. 40: Nine: Change to 9 for consistency.
Numbers were changed

L. 44: in 2012-2013: Pl. change to during 2013-2013, as monitoring was done in both years.
Changed to in 2012 and 2013

L. 77: Pl. add in China before the citation.
added

L. 104: decreased down to 107 km2: from what area? Is it about 500 km2?
Changed by "ranged seasonally between 489 in the WW season to 170 km$^2$ in the WD season during the course of the study."

L. 107: m3s-2: This is not a correct unit for discharge. Later you mentioned m3s-1 which is right.
Typo corrected

L. 123-125: besides the hydrology details which were already described in Guerin et l. (2016), it would be good if you can give depths of the stations also.
Hydrology and depth were given in Guerin et al (2016) as it was mandatory information for the understanding of the spatial variation. As we would have to give ranges, the addition of hydrology and depth details would impair the readability of the discussion without providing any substantial clarification or useful information..

L. 159: What is specific water discharge?
Done

L. 197: soils types: Pl. correct to soil types
Done

L. 199: details: Pl. use singular (detail) as above. And pl. make similar corrections elsewhere also.
Done

L. 199: Table 1 – what is interm. for?
Rewritten as follow ("interm.up" and "interm.down" samples, with interm standing for intermediate)

L. 213: One of the subsample: Pl. correct it as one of the subsamples (Pl. compare with the above two corrections).

done

L. 221: What is specific water discharge? What is Hum?
As those lines refer to the description of the soil static chamber, we do not understand the comment. Elsewhere in the MS, 'Hum' might refer to humidity.

L. 236: In Fig. 2, it would be better if the data are provided for the area classification followed in Fig. 8.
We are puzzled by this comment. Fig 2 reports particulate and dissolved inputs from the main pristine tributaries of the reservoir when Fig 8 reports GHG emissions from the reservoir itself and all impacted area.

L. 255 (also L. 638): This data has not been critically discussed.
The paper includes TIC, and TOC as they are needed for providing a carbon mass balance to identify the source of carbon fuelling $CO_2$ emissions, however the focus of the paper is on $CO_2$ emissions to the atmosphere. Furthermore, a detailed discussion on carbonate chemistry would require high precision pH data that we do not have to calculate equilibrium.

L. 259: This figure is illegible. The trends are not clearly seen due to the problem of scaling of the X-axis.
In order to be able to observe seasonal variations, the same scale is used whatever the season. The scale was adapted for each site to improve readability

L. 300: Are 70% and 56% (for 2011 and 2010 & 2012) annual average $O_2$ saturation values or seasonal values? Pl. clarify. Pl. modify text for better clarity.
"On average" was added for both cases, referring to annual calculation for 2011 and for the years 2010&2012

L. 301: the is a repetition.
removed

L. 325: From March to August: You have referred so far to seasons. Better be consistent and refer as WD and WW seasons.
March to August encompass the second half of the WD season and the first half of the WW season, therefore referring to season would not depict reality. Therefore, giving the precise months was here the most accurate way.

L. 332: space between five and fold.
Done

L. 338: Pl. change a to an.
Done

L. 337-340: Why was this? Pl. explain in Discussion.
It is already explained here L524-526. As found for CH4, the main factor influencing the spatial variability of $CO_2$ in the water column is the vertical mixing of the water column induced by the water intake located close to RES9 (Deshmukh et al., 2016;Guérin et al.,

2016). The design of the water intake enhances horizontal water current velocities and vertical mixing which lead to the transport of bottom waters to the surface. As a consequence, surface concentrations at RES9 were up to 30 times higher than at other stations in 2010 and 2011 (Figure 5b).

L. 342: Pl. use on instead of to.
Done

L. 344: Figure 5e: These are also the trends shown in 5c. Suggest removing. Suggest removing 5f also as this data is given in Table 3 (column 3).
Panel 5c provides average diffusive fluxes of $CO_2$ in mmol m-2 d-1 at the stations RES1-8 while the panel 5e shows total diffusive emissions at the stations RES1-8 + RES9 in $GgCO_2$ month-1 showing the relative importance of RES9 in the total diffusive emissions. Those data are not shown in table 3. Similarly, the panel f includes information from RES9 not given in table 3

L. 390: -32-33762: Pl. clarify the hyphen. The first hyphen seems to be a negative sign and the latter for range.
Done

L. 391: (not shown): The data can be included in Figure 5 as replacements for 5e and 5f to be deleted (see an earlier comment).
The figure 5 is about fluxes at the reservoir surface while the data on L391 are from the channel downstream of the powerhouse

L. 401: Fig. 6b: This data are included in Table 3 (column 5). Pl. remove.
What is the problem of citing in the text values included in a table?

L. 419: Where is Figure 6d?
Typo: fig 6b

L. 424-425: no bubbles was ever observed for depth higher than 16 m: Pl. delete text as this was given in Methods section.
Reworded as follow: The $CO_2$ content in the sampled bubbles was 0.29±0.37% (n=2334). On average, the $CO_2$ bubbling was 0.16 ± 0.24 mmol m-2 d-1 (0-2.8 mmol m-2 d-1) for depth shallower than 16m.

L. 434: stagnic property: Pl. explain briefly what a stagnic property is.
This is the classic term of pedology which is defined by the International soil classification system meaning that the soil was flooded. "stagnic properties: saturated with surface water (or intruding liquids), at least temporarily, long enough that reducing conditions occur"

L. 439-440: surface moisture ranging from 17.5 to 51.2% and temperature ranging from 18.1 to 34.2∘C (Table 2): For consistency, pl. change text as: surface moisture (17.5 - 51.2%) and temperature (18.1 -34.2∘C) (Table 2).
done

L. 443: This p value of 0.452 is not significant! Is it a typo?
Typo, 0.0452

L. 443-445: This sentence is not self-explanatory.
Rephrased as follow: Since we did not observe significant spatial variations related to topography, humidity or temperature that could have been considered for refine spatial and temporal extrapolation, we further consider the average of all fluxes that is 279±27 mmol m-2 d-1 throughout the years.

L. 449: could reach: For consistency, pl. change to reached.
Replaced by have reached

L. 450, 451: Pl. change changes to changed.
Done for reaches/reached

L. 454-455: Fig. 7 indicates that 2012 emissions were higher as July and August were also CO2-emitting. Pl. explain why under Discussion.
Modified as follow: Around 80-90% of the annual emissions occurred within 4-6 months of transition period between the WD and WW seasons (Figure 7) when the drawdown area surface is at its maximum

L. 473-475: This sentence is a repetition of the earlier sentence in content.
The two sentences were combined as follow: However, no $CO_2$ burst was observed at the beginning of the CD season evidencing that reservoir overturn has only a moderate impact on $CO_2$ emissions.

L. 475: This assumption is reinforce: Pl. correct to This assumption is reinforced
Done

L. 475: hot moments: When were those hot moments and why?
The CH4 emission dynamic depending on burst of emissions during overturn, and often called hot moments, is described in Guerin et al. (2016). Any detailed description on this phenomenon is beyond the scope of the manuscript under evaluation.

L. 477: the higher concentrations were observed: Pl. remove the definite article. Also, explain why.
Modified as follow: As observed in most tropical and subtropical reservoirs, the higher concentrations were observed during the warm seasons due to long residence time of water and warmer conditions enhancing CO2 build-up (Abril et al., 2005;Kemenes et al., 2011;Chanudet et al., 2011) whereas the lowest were found after reservoir overturn when the water outgassed (Chanudet et al., 2011).

L. 481: Pl. change the first of to a L. 482: change was observed nutrient concentrations: Correct to change was observed in nutrient concentrations.
done

L. 487-489: No, the quantity of autochthonous OM is not greater than phytoplankton primary production. Hence, there should be some other mechanism (source).

Inland waters are mostly heterotrophic which indicates that they "must receive significant inputs of organic carbon from adjacent ecosystems, assigning an important role to the lateral exchanges of carbon between land aquatic ecosystems (Duarte. and Prairie, 2005). Reservoirs are an extreme case since during the first years, most of the carbon is supposed to come from the flooded vegetation and soils (Abril et al., 2005, Guérin et al, 2008, Prairie et al., 2017 and this study)

Duarte, C. M. and Y. T. Prairie (2005). "Prevalence of heterotrophy and atmospheric CO2 emissions from aquatic ecosystems." Ecosystems 8(7): 862-870.
Prairie, Y. T., J. Alm, J. Beaulieu, N. Barros, T. Battin, J. Cole, P. del Giorgio, T. DelSontro, F. Guérin, A. Harby, J. Harrison, S. Mercier-Blais, D. Serça, S. Sobek and D. Vachon (2017). "Greenhouse Gas Emissions from Freshwater Reservoirs: What Does the Atmosphere See?" Ecosystems.,)

L. 498: older reservoir: Pl. change to older reservoirs.
done

Fig. 1: This figure is cluttered. The station codes are too long (and also not explained) and contribute to this clutter. What is the direction of river flow? What are NKT, TRC, DCH and XBF? The artificial channel is not marked properly in figure, and it is difficult to understand when mentioned e.g. in L. 530. Some terms included in legend e.g., Stream and downstream channel occur nowhere in text. Res 1 and downstream of reservoir – are they same? It should help the reader if you explained the provenance of different sampling stations in the Methods section, or as commented under Table 3.
Definition of all abbreviations are now given in the caption, the downstream channel is now better differentiated from rivers, arrows indicating the flow were added.

L. 549: For consistency, pl. change between the WD and the WW season (April – July).
done

L. 551: emissions factors: Pl. change to emission factors.
done

L. 552-557: This difference between CH4 (earlier work) and CO2 could be explored further.
Same explanation as before on the inclusion of CH4 in this article

L. 555: Table 3: Pl. give data separately for Res 9. This can be done by inserting a row after the header row for giving the stations included.
This is done in figure 5 already

L. 559: compare to most of the reservoirs: Pl. correct as compared to most of the reservoirs.
Done

L. 564-567: This sentence is a repetition from earlier discussed.

Compared to L541, the result of the overturn is added to the degassing at the water intake in order to explain the low downstream emission

L. 568: For consistency of tense, pl. change increase to increased.
Done

L. 568-569: This sentence is also a repetition (Pl. see the opening sentence of this Section!).
The opening sentence is on total emissions from the whole systems. Here, we are focusing on diffusive fluxes as clearly stated

L. 579: down to 7oC in air in March 2011: Was this given under Results?
It is included in the averages given in the site description

L. 620: were taking into: Pl. correct as were taken into
Done

L. 624: this study highlights: But this study is about CO2 only.
CH4 was removed

L. 688: Pl. correct algaes as algae
Done

L. 696: in a tropical reservoir: Pl. specify. Correct it as in the tropical NAM 2 reservoir.
This is the first study of its kind in a subtropical reservoir, therefore the statement is correct

L. 705: Pl. change all with different
Changed to "all known pathways"

L. 708-711: This sentence is redundant.
Redundant with which other sentence? As the conclusion is a place to put together all important findings of a study, the eventual redundancy is not, in our opinion, a problem.

L. 712: with time over the years: Pl. remove with time.
done

L. 713: represent: Pl. correct it to represents.
done

L. 715-717: But this is important only in the initial years after impoundment as evident in Fig. 8a. By the year 2013, the emissions have decreased significantly. Also, during the WW and particularly CD season, a seasonal shift of emissions happened and the reservoir emissions far surpassed the emissions from the drawdown area, thereby restoring the condition existed pre-power plant commissioning. Thus, the drawdown area and Reservoir have their own seasons when emissions peak – WD and the initial part of WW seasons in the former and the later part of WW season and CD season in the case of the latter. Pl. explain clearly under Discussion the result and why it is so.

The flooded biomass is the main source of carbon fuelling emissions, whatever the season and while permanently under water (high water level) or seasonally covered/uncovered (low water level). Emissions from the drawdown area obviously occur only at the low water level when the soils are not submerged.

L. 718-720: Although the % emissions from the drawdown area is 75% of total, in absolute terms, the emission (quantity) is same or perhaps less in 2013, as prior to commissioning.
Up to 75% was changed to 40-75%. Drawdown emissions is the only term which appears to be quite constant since the creation of the reservoir.

L. 729: footprint of the reservoir: What is footprint? Not discussed earlier under Discussion.
Reservoir footprint is the area of influence of the reservoir